# State-of-the-art global models underestimate impacts from climate extremes

Jacob Schewe ⓘ et al.#

Global impact models represent process-level understanding of how natural and human systems may be affected by climate change. Their projections are used in integrated assessments of climate change. Here we test, for the first time, systematically across many important systems, how well such impact models capture the impacts of extreme climate conditions. Using the 2003 European heat wave and drought as a historical analogue for comparable events in the future, we find that a majority of models underestimate the extremeness of impacts in important sectors such as agriculture, terrestrial ecosystems, and heat-related human mortality, while impacts on water resources and hydropower are over-estimated in some river basins; and the spread across models is often large. This has important implications for economic assessments of climate change impacts that rely on these models. It also means that societal risks from future extreme events may be greater than previously thought.

#A full list of authors and their affiliations appears at the end of the paper.

Estimation of the total damages caused by climate change requires a quantification of climate impacts across a large range of economic and societal sectors. These sectors include agriculture[1], water resources[2], energy supply and demand[3], human health[4] and ecosystem services[5]. There are approaches that integrate damages across sectors, such as the highly idealised damage functions used in integrated assessment modelling[6], but also more sophisticated, coupled economic modelling frameworks that combine individual sectoral models[7–9]. However, these approaches are centred on gradual changes in physical and biophysical indicators—such as crop yields or water resources—and largely ignore the impacts of extreme climate and weather events.

This is a serious research gap because such events cause enormous damages[10]. For addressing it, sectoral impact models must be able to credibly represent the impacts of extreme events. The goal of this paper is to test whether this is the case in the complex, process-based impact models that are routinely being applied in global-scale climate impact assessment[1–3]. While these models may be too costly to integrate them directly in cross-sectoral economic models, they are the benchmark for any simpler models. More generally, these complex impact models, in conjunction with global climate models, form the basis of much of our current knowledge about future global climate change impacts, as reflected in the Intergovernmental Panel on Climate Change reports, for instance. Whether they capture extreme events well is therefore a key concern even beyond the application in economic assessments.

And yet, it is not known how well the current suite of models can reproduce the multi-sectoral impacts of a given climatic extreme event. Global process-based impact models have been evaluated in terms of average quantities and sometimes in terms of inter-annual or intra-annual variability[11–14], but their performance under extreme conditions has rarely been tested at large spatial scale[15], and never—to our knowledge—in a multi-sector setting. And since events that are very rare today may become much more frequent in the future[16], testing for variability alone may not be enough.

Here, we choose the 2003 European heat wave and drought (EHWD) event as a test case. The EHWD was substantially stronger than previously observed events; it severely impacted several important sectors across a large geographical area, and its impacts are relatively well documented. We examine the impacts of the EHWD in a large ensemble of state-of-the-art impact models covering agriculture, water resources, terrestrial and marine ecosystems, energy, and human health, for the first time in a common modelling framework. For each of these sectors, we identify key observed impacts of the 2003 EHWD reported in the literature and/or recorded in public databases, and examine how closely the models—driven by observations-based climate data—reproduce those impacts. As a common impact metric, we choose the deviation of 2003 from the historical average, adjusted for long-term trends, and normalised by the historical standard deviation (except for human health; see Methods). We thereby circumvent potential biases in the baseline or the average inter-annual variability, and instead focus on the models' ability to pick out the anomalous 2003 event from the rest of the time series.

The Results section first provides a climatological analysis of the 2003 EHWD, and then presents impact model results for each sector. In the Discussion section, we summarise and evaluate our findings across sectors, and discuss their implications for integrated assessments of climate change impacts and for future model development.

## Results

**The 2003 European heat wave and drought**. The 2003 EHWD stretched over the entire summer, with large and persistent hot anomalies especially during June and August (Supplementary Fig. 1), and it extended across much of Western and Central Europe (Supplementary Fig. 2). The June–August average temperature anomalies (relative to 1961–1990) were extreme, reaching 2 °C (2 standard deviations ($\sigma$) above the mean) averaged over Europe and more than 5 °C regionally (more than $3\sigma$, and in some locations $5\sigma$)[17,18]. Due to both decreased precipitation and increased evapotranspiration, the high temperatures were accompanied by an intense drought, and the dry soils in turn amplified the heat wave[19,20].

The EHWD had numerous impacts on the environment, economy and human health[18]. In this paper, we focus on large-scale impact indicators which can be compared between observations and global-scale models (Fig. 1). In terms of these indicators, the summer of 2003 is characterised by anomalies of up to $5\sigma$ in observed data. In particular, southern Europe saw extreme reductions in ecosystem gross primary productivity (GPP) and large excess human mortality rates. Substantial relative reductions in crop yields, river flow and hydropower production were experienced across the different parts of Europe affected by the EHWD. The ability of impact models to capture these large anomalies is summarised in Fig. 1, and discussed in the following sections.

The EHWD was exceptional compared to the historic record, but given continued global warming, comparable events are anticipated to occur about every 10 years by the middle of the 21st century, depending on future greenhouse gas emissions[21–23]. Even when changes in the mean climate state are discounted—tantamount to assuming full adaptation to gradual climate change—relative deviations like the 2003 EHWD could still occur compared to the background climate state in the late 21st century[24]. Furthermore, the impacts of the 2003 event could have been much greater if spring conditions had been drier[19], e.g. similar to those that prevailed in 2011[25], which underlines the risk of even more-extreme events in the future. Being able to estimate the damages from such events is therefore crucial for assessments of future climate change impacts.

**River flow and water resources**. Due to the prolonged rainfall deficit (Supplementary Figs. 1–3), the summer of 2003 was anomalously dry in central Europe. This was visible in surface runoff and river discharge levels[26–28], which are indicators of renewable freshwater availability for ecosystems and human uses, such as irrigation or the cooling of thermal power plants[29]. In August 2003, discharge in both the Rhine and Elbe rivers reached record low levels[30]. Navigation was impeded on the German section of the Rhine on 37 days in 2003 due to low flow[31]. Satellite-based gravity measurements indicate an exceptional depletion of terrestrial water storage compared to 2002[32].

We examine monthly average river discharge at major gauging stations across Europe where continuous data since 1979 is available from the Global Runoff Data Centre, and compare this to an ensemble of global hydrological models (Methods). The observed data indicate negative anomalies of ~1.5–$2\sigma$ during August 2003 at five of the stations: in the Rhine (Lobith) and the Danube basin (Bratislava, Achleiten, and Inn at Passau-Ingling), as well as in the largest Swiss catchment, Aare (Figs. 1 and 2). These larger anomalies are reproduced closely by most model simulations, with a relatively small spread across the ensemble. The multi-model median is very close to the reported value at the Rhine, Tisza, Danube, and Aare stations. Conversely, many models overestimate the more moderate anomalies observed e.g.

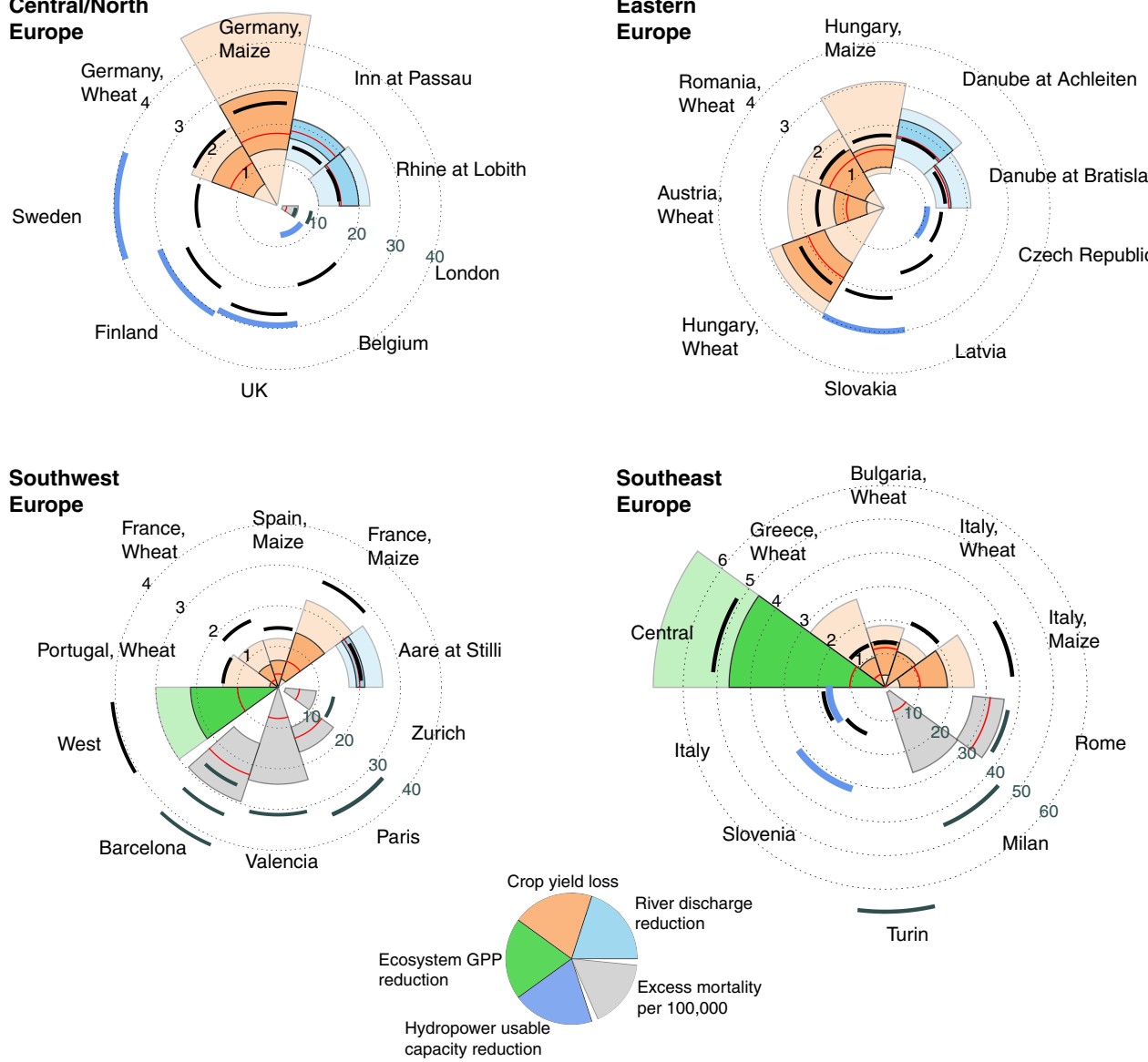

**Fig. 1** Multi-sector impacts of the 2003 EHWD. Black arcs represent observations, and colours represent model results. Units are standard deviations (black axis labels), except for human mortality which is given in excess deaths per 100,000 (grey axis labels). For river discharge, crop yields and ecosystem GPP, the thin red line marks the multi-model median; the dark-coloured segment marks the interquartile range; and the light-coloured segment marks the full range of model results. For hydropower, only one model is available which is marked by the thick blue arcs. For mortality, the red line and grey segments mark the median and the full range, respectively, across three climate forcing data sets and three different heat-mortality relations. Note the larger axis range for Southeast Europe. This figure only includes river discharge, crop yield, GPP and hydropower results for those locations where a negative anomaly larger than 1 standard deviation was observed. Figures 2–6 include further details on the data shown here, as well as additional data for locations with smaller or positive anomalies. The West and Central regions used for ecosystem GPP are defined in Methods

in the Elbe and Oder rivers, and the ensemble spread tends to be larger there (Fig. 2). Out of 12 stations, there are 5 stations where 75% or more of the models simulate a negative discharge anomaly larger than 1 standard deviation even though the observed anomaly is smaller. There is no station where the magnitude of the observed anomaly is underestimated by many models.

Results are similar in simulations that ignore the effects of human land use, dams and reservoirs, and water withdrawals (Supplementary Fig. 4), suggesting that present human modifications of the hydrological system do not substantially change the response of the system to this type of extreme event. This is consistent with the moderate effect of human interventions on

streamflow and drought conditions in central and western Europe reported in earlier studies[33,34]. Indeed, in the hydrological models that include human interventions, irrigation water demand would increase under drought, but this has little effect on river flow when the flow is already low and limits the amount available for withdrawal (Supplementary Fig. 5).

Model performance differs little between June, July and August (Supplementary Fig. 6). While there is some variation in which stations are matched best in each month, in those cases where there is a large anomaly (such as in the Danube in July) the model ensemble tends to reproduce that anomaly closely. Results are also insensitive to the choice of climate forcing data set

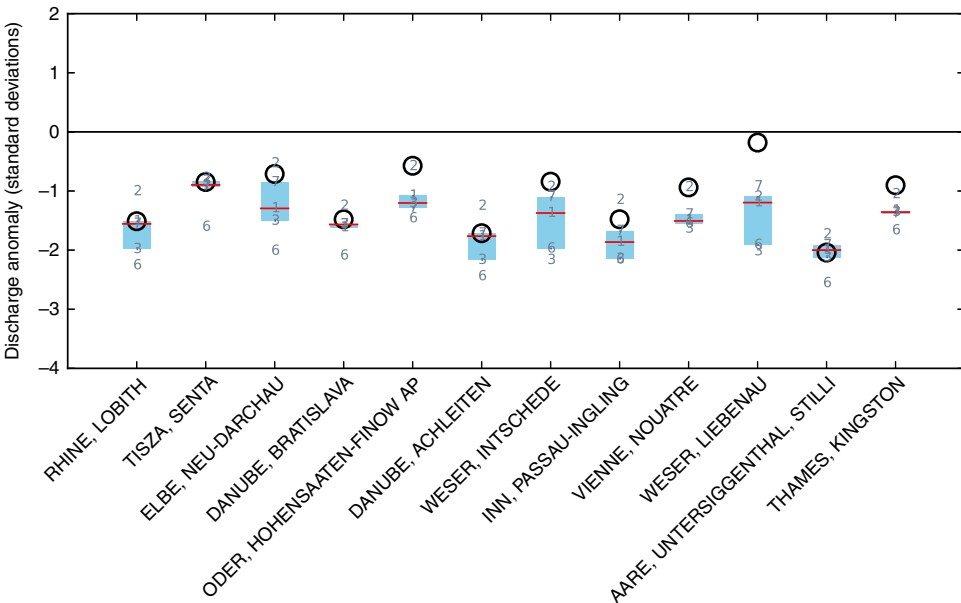

**Fig. 2** August average river discharge anomalies in 2003. Black circles are observed (GRDC) data. Grey numbers are the global hydrological models (see Methods); red lines indicate the median, and blue boxes the interquartile range, of the model ensemble. Stations are ordered by catchment size; the smallest catchment (Thames river at Kingston) has an area of about 10,000 km$^2$, which corresponds to the size of four model grid cells

(Supplementary Fig. 7). Our results are consistent with recent studies attesting a relatively good performance of global hydrological models for discharge and runoff variability in European catchments, compared to other regions[12,35]. At the same time, the tendency towards false-alarms is in agreement with a dry bias induced by the models' potential evaporation schemes[36]. The fact that one of our models (number 6) appears consistently at the dry end of the ensemble may be because it applies a temperature-based evaporation scheme (Hamon) that has been shown to induce a particularly large bias when applied outside its calibration range[36].

**Agriculture**. Agricultural summer crop yields were poor in 2003 due to the combined effect of the drought and the excessive temperatures recorded since June[18,37]. Low harvests incurred an estimated €13 billion of uninsured losses in the European Union (EU)[38]. At the EU level, maize and wheat were the most damaged crops, with production shortfalls of 21 and 11%, respectively[18]. We examine maize and wheat crop yields simulated by an ensemble of 12 global gridded crop models, and compare them with data reported by FAOSTAT (obtained from http://faostat.fao.org/site/567/default.aspx on 30 August 2016) for central and southern European countries with continuous data since 1979. Consistent with previous studies[39] and early assessments by the COPA-COGECA agricultural association[40] (Supplementary Table 2), we find the 2003 EHWD was associated with large negative yield anomalies in the FAOSTAT data set in France, Germany and Italy for both maize and wheat; as well as in Spain for maize, and in Austria and Portugal for wheat (Fig. 3, black circles). Additionally, we find substantial yield reductions of one standard deviation or more for both crops in Hungary, and for wheat in Bulgaria, Greece and Romania. Hungary and Romania in particular are among the EU's largest maize producers.

The ability of the crop model ensemble to reproduce these impacts is mixed (Fig. 3, boxes and numbers). Out of 5 countries with an observed negative maize yield anomaly larger than

1 standard deviation, there are 3 (4) countries where all models (more than 75% of models) underestimate that anomaly. For wheat, the numbers are 2 (7) out of 9 countries. The best-matching models differ from country to country, in line with a previous evaluation of these models[11] which also found mixed skill in reproducing overall inter-annual yield variability. There is broad agreement across models on the sign of the anomaly in most of the strongly impacted countries; except for wheat in Italy, France, Portugal and Greece, where nearly half of the models show a positive anomaly. In terms of magnitude, the large anomalies in France and Italy are underestimated by the entire ensemble for both crops. Generally, the agreement is somewhat better in Middle and Eastern European countries, such as Hungary, Bulgaria or Poland, compared to Western Europe. These general results are robust against changing the climate forcing data set, although the yield loss in Italy is better reproduced with an alternative data set (Supplementary Fig. 8). We also note that COPA-COGECA report a decline in maize yields in Austria by about 10% between 2002 and 2003, which is not reported in the FAOSTAT data (Supplementary Table 2); thus, the real observed value in Fig. 3 may be closer to the model ensemble mean than the FAOSTAT value shown by the black circle.

The poorer performance in Southwestern versus Eastern European countries may be due to more widespread irrigation in the former. The crop models assume full irrigation in irrigated areas and do not account for potential limitations in water availability due to drought, which induces an overestimation of irrigated yields, and thus biases total yields in countries with much irrigation, such as France or Italy (Supplementary Fig. 9). This is in line with a recent study showing that extreme heat leads to strong declines in maize yield in both observations and models only under rainfed conditions[15]. Moreover, vernalisation of winter crops (the requirement of cold temperatures for flowering) is a known problem in crop models[41], and the positive sign of the simulated wheat yield anomalies in southern Europe in many models may be due to winter wheat flowering, maturing and being harvested early and thus escaping the heat wave[42]. Finally, as is the case with the other sectors, the crop model simulations were not specifically designed to

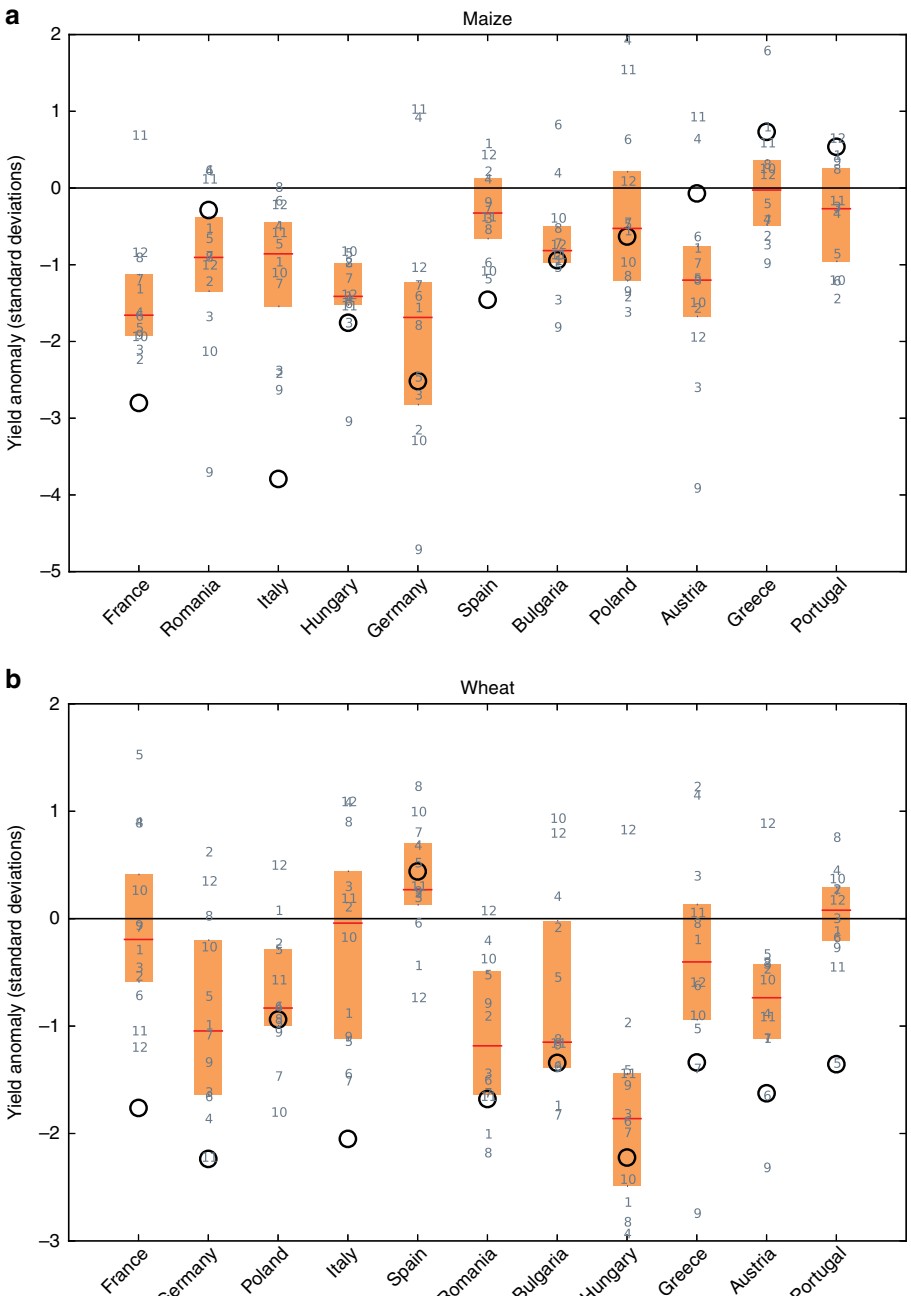

**Fig. 3** Crop yield anomalies in 2003. **a** Maize, **b** wheat. Black circles are observed data from FAOSTAT. Grey numbers are the 12 global gridded crop models; red lines indicate the median, and orange boxes the interquartile range, of the model ensemble. Countries are ordered by their total production (http://ec.europa.eu/eurostat/web/agriculture/data/database) in 2010, decreasing from left to right

reproduce 2003 yields, and used standard management and phenology settings; for instance, fixed planting dates and a constant number of heat units required for a crop to mature.

**Terrestrial ecosystems**. The summer of 2003 saw a major reduction in overall GPP in parts of Europe, based on vegetation model results[39,43] and remote sensing greenness index data[44,45]. The largest anomaly was situated in France, but negative anomalies were also observed in northern Spain, Italy, and parts of the Balkan and Germany (Supplementary Fig. 10).

We evaluate an ensemble of global vegetation models over two overlapping rectangular regions (Fig. 4a): one (West) comprising France and northern Spain, where the centre of the negative GPP

anomaly was located; and one (Central) including Germany, central Italy, and the western Balkans, which also saw large negative anomalies, but also including the Alps which saw a positive anomaly[46]. As observational benchmark we use MODIS[47,48] GPP estimates derived from remotely sensed absorbed fraction of photosynthetically active radiation (fAPAR; Methods). We find negative anomalies of 4 to 5σ in MODIS-derived GPP during 2003 (Fig. 4b, black circles), consistent with previous analyses[45].

The magnitude of the observed anomaly is underestimated by 75% of models in the Central region, and by all models in the West region. In both regions the model ensemble spreads over a large range that also includes positive anomalies and near-

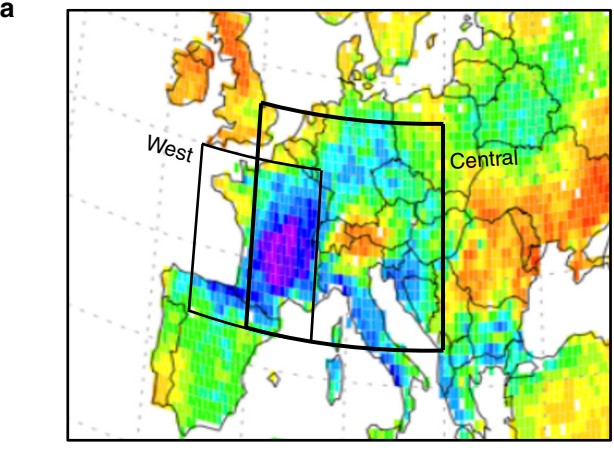

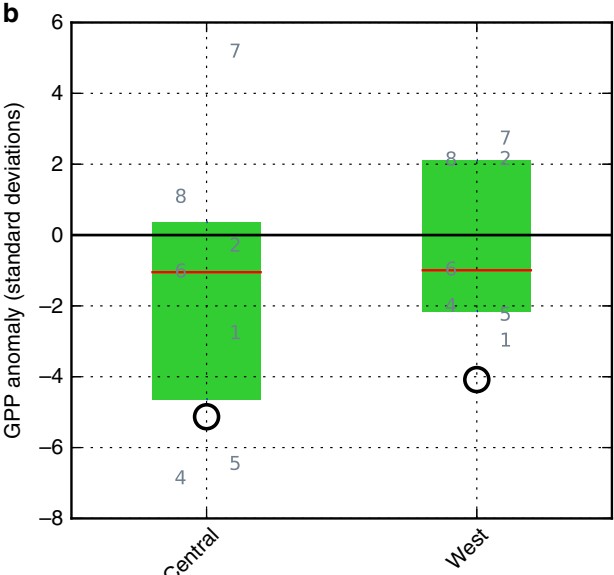

**Fig. 4** Summer (June–August) gross primary productivity (GPP) anomaly in 2003. **a** Outlines of the West and Central regions, overlaid on a map of the observed GPP anomaly; see Supplementary Fig. 10 for a more detailed version of this map. **b** Regionally averaged anomaly. Black circles show MODIS remote-sensing-derived estimates. Grey numbers are the global vegetation models; numbers offset to the left represent models that were run without considering any human influence except climate change, while numbers offset to the right represent models that accounted for historical land-use patterns and, in some cases, anthropogenic water withdrawal. Red lines indicate the median, and green boxes the interquartile range, of the model ensemble. One model (no. 4) did not report total GPP but only GPP for individual plant functional types (PFTs); for this model, we show the sum of the four dominant PFTs in the relevant region

zero values (Fig. 4b, numbers). There is no systematic difference between models that include and exclude human-induced changes to the natural vegetation pattern (land use change). The estimates are largely insensitive to the choice of climate forcing data set (Supplementary Fig. 11), despite substantial differences in annual shortwave radiation between data sets[49].

Our finding of a smaller sensitivity of GPP to drought in models than in observations is consistent with a previous multi-model study which, however, assessed a shorter time span[45]. Inspecting individual models' GPP time series (Supplementary Figs. 12 and 13), we note that those models that most closely reproduce the 2003 relative anomaly also exhibit a pronounced

positive anomaly in 2007 and 2008, consistent with the MODIS estimates. The large spread across models for the reduction of GPP in the 2003 EHWD is despite a relatively high spatial correlation with MODIS-derived GPP globally in all these models[49].

Previous studies[50,51] have suggested that many global vegetation models do not capture the vegetation response to drought, that is, extremely low soil moisture and air humidity, especially for unprecedented or long-lasting cases. While the models adopt mechanistic, or process-based, approaches, most models include empirical parameterisations for temperature and moisture response functions, which may be less reliable outside the historical range of variability for which they have been calibrated. Moreover, not all ecosystem models account for heat stress effects on plant photosynthesis and water stress effects on respiration[52], which might explain the underestimation of the EHWD's impact by many models. Other possible reasons include the lack of differential species response to drought[53], the lack of deep rooting access to water for forests, irrigation and more generally a poor description of the phenology of cultivated vegetation[51].

**Energy**. Low water availability and high water temperatures during the 2003 EHWD affected electricity supply due to declines in the output of hydropower plants, and impaired operation of thermoelectric power plants that require freshwater for cooling. Across Europe, more than 30 nuclear power plant units had to reduce their production in summer 2003 because of constraints for cooling water uses[54,55]. Together with hydroelectric power restrictions, this led to a doubling in electricity spot market prices compared to the previous summer[30]. Even wind power potential was at record low levels in several countries (Methods).

We analyse model simulations of hydropower plants' usable capacity, i.e. the maximum available output at a given point in time (Methods). In the absence of observations of usable capacity, we compare these simulations to reported annual hydropower generation from the global database of the U.S. Energy Information Administration (EIA); while noting that a plant's actual power generation can be lower than its usable capacity depending on management. We exclude countries with less than 20 plants or less than 100 MW total installed capacity. The observed data show substantial negative anomalies of around 1.5–2.5$\sigma$ in 2003 for a number of countries, among them Italy which is one of the largest hydropower producers in Europe (Fig. 5). Sweden, Finland, and the UK registered large negative anomalies, too; consistent with the observation that northern Scandinavia and the British Isles experienced dry conditions in summer 2003, even though they were not at the centre of the heat wave. At the same time, Spain, Portugal, and Greece registered above-average power generation; consistent with near-average summer precipitation and slightly above-average precipitation during the rest of the year (Supplementary Figs. 2–3).

The simulated anomalies are very close to the observed ones for Italy and the UK, relatively close for Finland and the Czech Republic, and also closely reproduce the positive deviation in Greece. Large observed negative anomalies in Latvia and Belgium are not reproduced. For the remaining countries, the model mostly overestimates the negative anomalies; underestimates the positive anomalies; and simulates below-average usable capacity for some countries that saw near-average power generation, such as Switzerland and Germany. The latter may be because the underlying hydrological model does not capture enhanced glacial melting during the heat wave, which could offset runoff deficits downstream. In total, out of 9 countries with an observed negative hydropower generation anomaly larger than 1 standard deviation, there are 3 countries where the usable capacity model

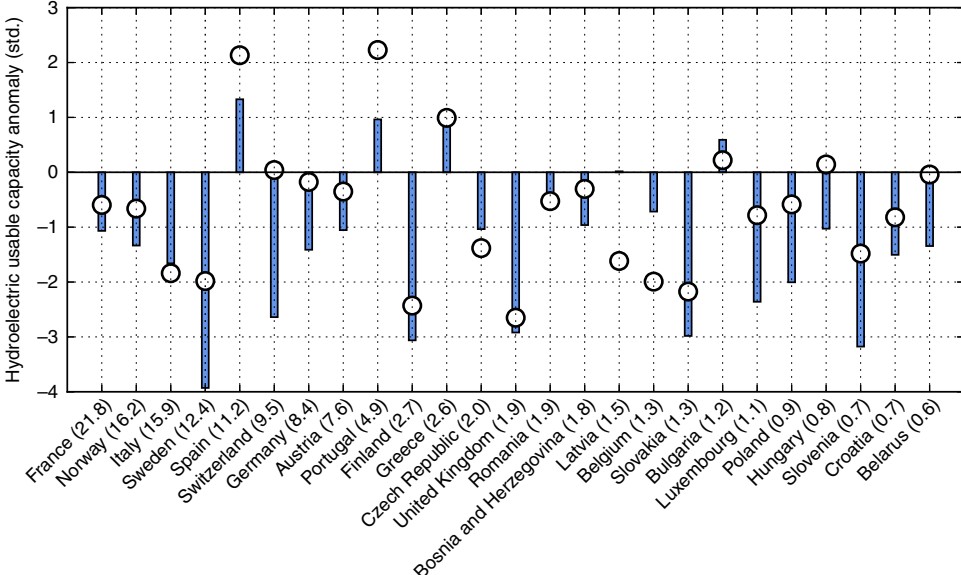

**Fig. 5** Hydropower anomalies in 2003. Black circles show anomalies in annual hydropower generation reported by EIA (Methods). Blue bars show anomalies in simulated annual hydroelectric usable capacity. Countries are ordered by installed capacity (in GW) as included in the model, indicated in parentheses

underestimates the magnitude of that anomaly. On the other hand, out of 25 countries, there are 10 countries where the model simulates a negative discharge anomaly larger than 1 standard deviation even though the observed anomaly is smaller in magnitude.

A previous evaluation of the hydropower model, using the same climate input, found relatively low root mean squared error in southern European countries, but larger values in some north European countries, with particularly poor performance in Latvia[56]. Our analysis broadly confirms this regional pattern, while focusing explicitly on a drought event. We note the complexity in comparing actual hydropower generation with hydropower usable capacity, which is simulated by the model. In particular, annual power generation is strongly affected by power demand, which was above-average in the winter of 2003 due to low temperatures. The hydropower model focuses on the physical impacts of changes in water resources on usable power plant capacity; economic feedbacks of the assessed water constraints and related management or adaption options (e.g. on energy prices and the supply-demand portfolio) are not modelled. More importantly, no monthly observational data are available, so that we can only compare annual averages between model and observations. Simulated summer-only (June–August) anomalies are generally larger than for the annual average, and show a similar relative distribution across countries, which suggests that the anomalies seen in the annual average are largely due to the EHWD (Supplementary Fig. 14).

**Human health**. The 2003 EHWD caused a high human death toll mainly due to circulatory- and respiratory-related causes, especially among the elderly population[57]. As many as 70,000 excess deaths were estimated for Europe, and 14,800 for France alone[58,59]. We apply a set of city-specific statistical models that describe a linear relationship between maximum daily apparent temperature ($AT_{max}$) and daily mortality (see Methods). We calculate $AT_{max}$ from daily mean and maximum temperatures and relative humidity, from the same observations-based climate data set used in the other sectors above, as well as two additional data sets (GSWP3 and PGFv2).

There is no comprehensive and consistent database of city-scale excess mortality to compare our estimates to. Instead, we compile results from earlier studies that specifically estimated the effect of the 2003 EHWD on natural mortality in European cities (Supplementary Table 3). These are based on actual mortality data reported for the 2003 summer e.g. by hospitals and city authorities, and are the closest available analogue to direct observations. In contrast, the models we applied[60] were trained on mortality and climate data from 1990–2000, and our estimates for 2003 are then derived based on the estimated model, reported baseline mortality rates, and the climate forcing.

We find that our statistical estimates are consistent with the range of literature values for Barcelona, London, and Rome, while they are substantially below the observed value(s) for Paris, Milan and Turin (Fig. 6). For Turin, our estimate is zero for all three climate forcing data sets, while an analysis[61,62] based on observed mortality found more than 60 excess deaths per 100,000 people occurred during the 2003 EHWD. For Valencia and Zurich, our models also underestimate observed values, although they come somewhat close when using the upper 95% confidence level for the slope of the temperature-mortality relation. Altogether, our models substantially underestimate the magnitude of the impact in 5 out of 8 cities for which excess mortality was observed. To our knowledge, this is the first independent evaluation of the performance of these models outside their calibration period.

Potential reasons for the discrepancies between our model estimates and observed mortality include compounding effects that the statistical models do not account for, including: the added effect of heat wave duration, the urban heat island effect, and air pollution (both with particulate matter and ozone); as well as the difference in spatial resolution between the climate data used to train the models (local airport weather stations) and the 0.5°-resolution forcing data sets (see Methods for an extended discussion).

Interestingly, while results are similar across the different climate forcing data sets for most of the cities, we find a strong dependence on the climate forcing for Milan. Unlike impacts on water resources or plant growth, which integrate the weather

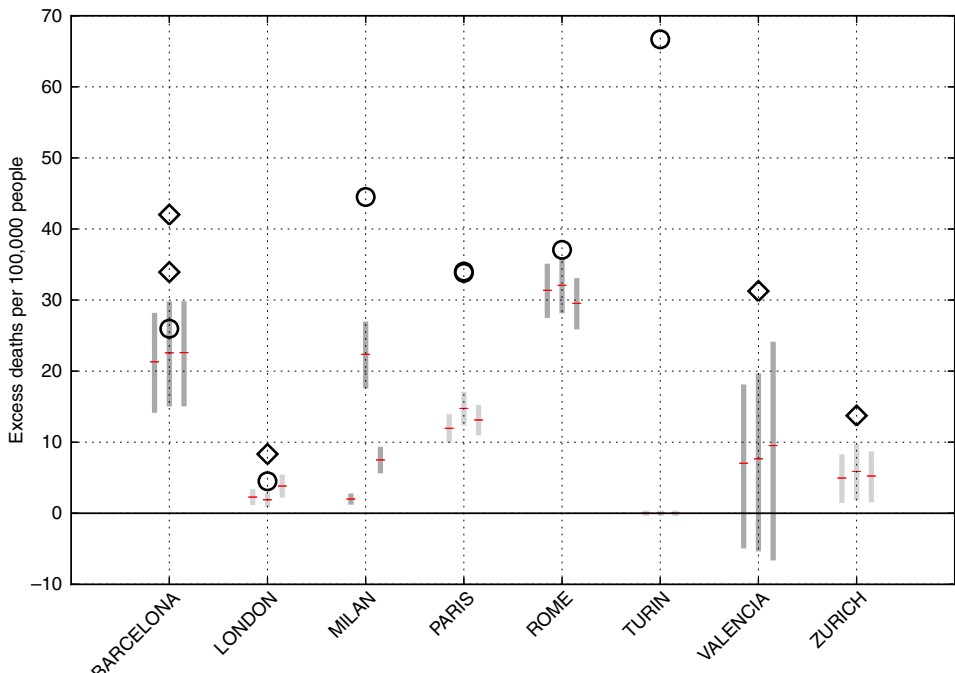

**Fig. 6** City-specific estimates of the excess mortality attributable to the 2003 EHWD. Black symbols are observed estimates from the literature (Supplementary Table 3; note two very similar estimates for Paris). Circles denote studies that have reported both the number of excess deaths and the baseline population; diamonds denote studies which have only reported the number of excess deaths and where we have used the corresponding city population reported for 2003 in official statistics as a baseline. Grey bars and red lines are results from this study, for three different climate forcing datasets (left: GSWP3, middle: PGFv2, right: WFDEI). The bars indicate the results obtained by using the lower and upper 95% confidence intervals for the linear exposure response function slopes from ref. [60], and the red line indicates the result obtained using the central estimate for the slope

effect over timespans on the order of weeks to months and over larger areas, excess mortality can result from just a few days of extreme heat in a particular location (city). Thus, small-scale deviations between the different climate forcing datasets or, for that matter, between a gridded reanalysis dataset and the local observations, can lead to large differences in the mortality outcome.

**Marine ecosystems.** The 2003 EHWD also produced anomalously high sea surface temperatures (SST) in the seas around Europe, most pronounced in the north-western Mediterranean Sea (Supplementary Fig. 15). This surface warming affected subsurface temperatures, vertical mixing, and currents, with potential effects on marine ecosystems and, by extension, on fisheries[63]. Indeed, several impacts, positive as well as negative, on local and regional marine ecosystems have been reported in the literature, including a die-off of *Posidonia oceanica* seagrass habitats[64] and rocky benthic communities[65]; and effects on the reproduction of some fish species like Bluefin tuna[66], anchovy and round sardinella[67]. However, no direct impacts on commercial fisheries are reported in the literature due to the EHWD, despite evidence for longer-term effects of warming on fisheries landings[68].

We simulated total consumer biomass at a 0.5° horizontal resolution with an ensemble of four marine ecosystem models[69]. As there is no widely accepted long-term database of fish biomass appropriate to the scales relevant to the EHWD, to gauge the impact of the event on marine ecosystems and fisheries, we compare relative change in reconstructed catch[70] to the relative change in simulated consumer biomass, taking the assumption that harvest is a reasonable indicator of biomass. We find no discernible impact of the 2003 EHWD on fish biomass in any of

the models, nor in the reconstructed catch data (Supplementary Fig. 16). In a few cases, models show a drop in 2004, which may be a lagged effect of the EHWD, but this is hard to verify in the absence of a longer time series, and not supported by the reconstructed catch data.

Nonetheless, the ocean reanalysis dataset used to force the marine ecosystem models does capture the 2003 anomalous warming in the Mediterranean and North Sea with a similar magnitude as in the observed data (Supplementary Fig. 17). It also shows a local maximum in plankton net primary productivity (NPP) in 2003 in the Mediterranean and the North Sea, and captures an observed phytoplankton abundance peak in the English Channel[71] (Supplementary Fig. 18). At the same time, simulated NPP exhibits large year-to-year variability, and the 2003 value is generally less exceptional in a long-term context than for temperature (Supplementary Fig. 19).

Thus, literature, data and models appear to agree that the 2003 EHWD affected physical ocean parameters as well as plants and planktonic NPP, but that the signal did not propagate to higher trophic levels on a large scale immediately. This may be because the event was too brief or too weak to cause a significant change in upper trophic levels. Another reason may be that temporary warming can act on fish biomass in two opposite ways: on the one hand, higher temperatures tend to increase the amount of food that fish need to maintain growth rates, as well as their mortality due to predation, disease or senescence. On the other hand, short-term warming increases phytoplankton growth, which has a positive effect on fish biomass. The net effect may thus be small. Further evaluation of marine ecosystem model simulations resolving individual species, and/or forced with higher-resolution climate data e.g. from regional climate models, might illuminate whether the models agree with the data for the right reasons; and longer simulation runs would be needed to account

for any lagged effects beyond 2004. Nonetheless, our results suggest that this modelling framework is applicable to extreme events like the 2003 EHWD in a similar fashion as in the terrestrial impact sectors.

## Discussion

Our study is unique in assessing the ability of several state-of-the-art impact models, for multiple sectors, to replicate the impacts from a recent, high-impact climatic extreme event, the 2003 EHWD. When impacts occur simultaneously in different sectors, it is important that models replicate not only the average variability of sectoral impact indicators, but the actual timing and magnitude of particular events. The rationale of our study was therefore to approach the assessment of model ability differently —not through a classical evaluation of impact statistics over several decades[11,14,35]—but instead to answer a what-if question: what if these models are used to predict the impacts of a single extreme event like the EHWD?.

The answer, according to our results, is that the total impacts of such an event would likely be seriously underestimated. This is despite some encouraging results in individual sectors. For instance, the observed low water levels in the Rhine, Danube, and Aare rivers are matched very well by the median across our hydrological models, and the spread across models is moderate. Furthermore, in most countries where a substantial reduction in hydropower generation is observed, our hydropower model also simulates a substantial reduction in usable capacity. With respect to human health, our statistical models yield heat-related excess mortality rates of a similar magnitude seen in some observed estimates for Barcelona, London and Rome. We also find that generally the choice of climate forcing data set is not a major source of uncertainty.

Importantly though, severe impacts on agriculture and ecosystem productivity are underestimated by a large margin by most of the sector models. None or few of the crop models (depending on the climate forcing) reproduce the extremeness of the negative maize yield anomalies, of three and four standard deviations, respectively, in France and Italy, which are both among the largest maize producers in Europe. The crop model ensemble also systematically under-predicts the severity of the EHWD's impacts for wheat in Western Europe, implying that increasing temperatures and more frequent heatwaves due to climate change could have substantially more severe impacts in the region than typically assumed. Similar results are found for natural ecosystem primary productivity: only a few of the global vegetation models come close to the enormous negative anomaly seen in remote sensing-derived GPP in Western Europe, and some models even display a positive anomaly. For both the crop and GPP models, neither the multi-model mean nor any individual model reliably reproduce observations. The mortality models estimate only half of the observed rate of heat-related excess mortality in Paris, which saw by far the most fatalities of all cities.

Taking all sectors together, and across all countries, cities, regions and rivers studied, severe negative impacts (deviation of more than $1\sigma$, or more than 10 excess deaths per 100,000 people) were observed in 38 cases (Fig. 1). In 20 of them, the severity of impacts is underestimated by the respective models, counting in those countries where 75% or more, but not all, of the crop models underestimate yield impacts. Leaving out those countries, we are left with 14 cases where the entirety of models underestimates the observed impacts. This summary statistic is necessarily crude, but highlights the magnitude of the problem.

While many of the largest impacts of the EHWD are thus underestimated by the sectoral model ensembles, we also find a number of "false alarms", mainly for water resources and hydropower. Hydrological models show flow reductions of up to

$2\sigma$ in the Weser, Elbe and Oder rivers, where no substantial anomalies were observed. The hydropower model shows large reductions in usable capacity in Switzerland, Poland and Germany, while reported hydropower generation was near average in all these countries. On a regional level, the underestimation of impacts across sectors is particularly evident in southern Europe; while in northern and eastern Europe, impacts on crop yields and river flow are matched relatively well, and some hydropower impacts are even greatly overestimated (Fig. 1).

Whilst our study is the most comprehensive systematic comparison to date between simulated and observed impacts for an extreme climatic event, it is not exhaustive in the treatment of different sectors, e.g. we did not include livestock production, vector-borne diseases or forest fires[18]. Assembling an even larger multi-disciplinary team could have partly addressed this. For some sectors suitable observational data for model comparison were unavailable (nuclear power generation). In all sectors, we have focused on large-scale impact indicators—e.g. average monthly discharge or country-level crop yield—which are close to what the models are designed to represent, and which we assume to reflect the total societal costs and key environmental impacts of the 2003 EHWD to a large extent. Smaller-scale characteristics of some of the impacts (e.g. minimum daily river flow or maximum daily excess mortality) could have important implications not captured by the more aggregate indicators, for example the overburdening of critical infrastructure for a short time. Representing such processes accurately will be a substantial challenge for the relatively large-scale models applied here.

We have investigated a single climate event, but our results are relevant beyond the case of the 2003 EHWD. In 2015 and 2018, central Europe was again hit by extremely hot and dry summer conditions, and projected trends in temperature and precipitation suggest that such events will likely occur more frequently in the future[72]. While each event is unique, similar events produce similar biophysical impacts: future drought-heat wave events will again lead to low river flows, reduce ecosystem productivity, and damage crop yields. As long as the model shortcomings exposed by our analysis persist, they will likely lead to biased estimates of extreme event impacts.

These shortcomings are related to the representation of both natural processes and human management in the models. For instance, human adaptation (or the lack thereof) can affect the severity of impacts for a given extreme event[73]. Indeed, excess human mortality during the 2006 heatwave in Europe was much lower than 2003, possibly because of better preparedness[57]. This might go some way towards explaining the underestimation of observed impacts by the heat-related mortality models: the 2003 EHWD was outside the climatological range that the models were trained on, and the range that the health system was prepared for. In process-based models, the capacity to respond to extreme climate conditions can be overestimated if resource constraints are not accounted for. An example is the lack of constraints on irrigation water in our crop model simulations. For instance, streamflow in the Vienne basin, which contains large shares of irrigated maize cropland[74], was anomalously low in summer 2003 (Fig. 2); and indeed, the maize yield anomaly in France is simulated more realistically when irrigation is switched off, than in the default model setup with irrigation (Supplementary Fig. 9). Accounting for such cross-sectoral interactions[75] and reflecting changing levels of societal adaptation in models will be critical for more accurate impact and damage estimates in the future.

Important natural processes such as heat stress effects on ecosystem photosynthesis are also missing in some models, while other processes such as potential evapotranspiration are parameterised through empirical relationships that may not hold outside past ranges of climatic variability[36]. Such shortcomings

will affect impact assessments of other climatic events, too. In addition, there are known issues that have not emerged in our study, but may prove problematic when assessing the impacts of extreme events of different types, or outside Europe. For instance, the global ecosystem models do not consider the potentially amplifying interactions between climate change and forest disturbances, such as wind storms[76]; and some global hydrological models struggle to capture the magnitude and timing of processes such as transmission loss and snowmelt accurately[35]. Finally, our models may perform worse in parts of the developing world where constraints from observational data are more limited.

Our results emphasise the importance of considering the uncertainties of current biophysical impact estimates. Modelling studies of future climate change impacts on mean conditions in different sectors[1,2,77,78] have shown that the uncertainties of biophysical impact estimates are of a similar magnitude as climate modelling uncertainty. A common practice has since been to use multi-model ensembles rather than single impact models, as is standard in climate modelling. We have shown that for an extreme event, even an entire ensemble of impact models can sometimes completely miss observations. This urgently calls for a shift in the agenda of model development and evaluation away from mean conditions towards extremes. In the meantime, a precautionary interpretation of impact estimates is recommended, where the most pessimistic model in an ensemble must be considered just as plausible as those in the centre of the distribution.

For quantifying the ultimate impacts of climate change on different aspects of the economy, or parts of society, different metrics need to be applied beyond the (mainly) biophysical indicators studied here; e.g., macro-economic damages, public health burdens, or effects on societal equality and political stability. However, any such analysis needs to rest on a firm assessment of the biophysical impact, and will be unreliable if the biophysical impact is not captured accurately. Our study thus provides important context to any subsequent analysis of economic and societal impacts.

## Methods

**Climate forcing data**. The primary climate forcing dataset used in this study is the latest WATCH meteorological forcing data set (WFDEI)[79], which is based on ERA-Interim reanalysis data. Two other data sets are used for comparison, for some sectors: the Global Soil Wetness Project phase 3 forcing data set (GSWP3; http://hydro.iis.u-tokyo.ac.jp/GSWP3/)[80], based on the 20th century reanalysis (20CR); and the Princeton Global Forcing data set version 2 (PGFv2)[81], based on NCEP/NCAR Reanalysis. The magnitude and spatio-temporal pattern of the 2003 summer temperature and precipitation anomaly is similar in all three data sets (Supplementary Figs. 1 and 2), and in agreement with other reanalysis products[17,18].

**Model ensemble**. Our multi-sectoral ensemble of impact models (Supplementary Table 1) includes 7 global hydrological models, 12 global gridded crop models, and 8 global vegetation models. In addition, we analysed the impacts on hydropower production as well as heat-related mortality, using single models; and potential impacts on fisheries using four marine ecosystem models. All terrestrial model simulations were driven by WFDEI climate forcing for the period 1979–2010, and followed a common protocol (ISIMIP2a, www.isimip.org), allowing us to compare models within a sector (such as agriculture) and to apply a common method of analysis across sectors. For some sectors, additional simulations were conducted, driven with PGFv2 or GSWP3 climate forcing. In those cases, ensembles are slightly different for the different climate forcings because not all models were run with all climate data sets. In Figs. 2–4, as well as the corresponding supplementary figures, models are uniquely identified by numerals (see Supplementary Table 1).

In contrast to the terrestrial models, the marine ecosystem models need not only atmospheric but also oceanic forcing variables, and are therefore driven by a reanalysis-based climate model run. While thus not completely consistent with the other sectors, these simulations demonstrate how our analysis framework can be expanded to the marine realm.

Importantly, the model simulations were not specifically designed to capture a particular event. Models were set up for best overall performance, and parameters were kept constant throughout the simulation period, with climate as the only source of annual-scale variations (except that the hydrological simulations account

for the construction of dams and reservoirs over time). This is similar to simulations under future climate change scenarios, where e.g. agricultural management choices in any particular year are not known. Therefore, if models are found to accurately reproduce the impacts of the 2003 EHWD using the present set-up, this increases confidence in their ability to quantify the impacts of future extreme events represented in climate projections.

**Impact metric**. In all sectors except human health, we measure impacts as a given variable's deviation in 2003 from the historical average, adjusted for long-term trends, and normalised by the historical standard deviation (see below for more details). That is, we measure the extremeness of the event in a given variable, not the absolute magnitude of change in that variable. If a model correctly simulated the absolute magnitude of change in a variable in 2003, but simulated the same magnitude of change every other year, then the agreement in 2003 may not be for the right reasons, and there would be little confidence in the model's response to extreme events in future climate scenarios. In the paper we use the term "impact" to refer to the relative deviation, or extremeness, of an indicator, unless otherwise stated. An exception is heat-related excess mortality, which is already defined as a deviation from the expected baseline mortality.

**Water resources**. Monthly discharge data was obtained from the Global Runoff Data Centre (GRDC, http://www.bafg.de/GRDC/EN/02_srvcs/21_tmsrs/riverdischarge_node.html) for gauging stations along major rivers in the geographic region that was affected by the 2003 EHWD. Only stations with catchments of ~10,000 km$^2$ (equivalent to ~4 model grid cells) or larger were selected. Stations also had to have continuous monthly data available from GRDC for the whole period 1979–2008. Simulated daily discharge data was first averaged to monthly values. Then, for both simulated and reported discharge, a single month of interest was extracted (e.g., August), resulting in a yearly series of values for that month over 1979–2008. From this series, a linear trend was removed, and the standard deviation calculated. The 2003 relative discharge anomaly was then calculated as the difference between 2003 and the 1979–2008 average, divided by the standard deviation (with 2003 excluded from the calculation of the standard deviation). We note that none of the models consider changes in glacial meltwater runoff, which was an important factor during the 2003 EHWD in Alpine catchments[26].

**Agriculture**. We analyse crop model simulations covering the period 1980–2009 under the "default" setting of the ISIMIP2a simulation protocol, where modelling teams individually determined how sowing dates, harvesting dates, fertiliser application rates and crop varieties are parameterised in their model, with the objective of best overall performance for the simulation period. Total crop yield in each grid cell was calculated as the weighted sum of irrigated and rainfed yields, with the weights being the respective cell fractions under irrigated and rainfed cultivation around the year 2000, as reported in the MIRCA2000 data set[82]. The 2003 crop yield anomaly was calculated from the simulated and observed series of annual harvests, analogously to the discharge anomaly, as described above; except that rather than a linear trend, a 7-year moving average was removed from the FAOSTAT reported crop yields, in order to account for non-linear changes induced e.g. by technological and management changes. Such factors are absent in the crop model simulations, but we applied the same de-trending method to the simulation data for consistency. The size of the moving average window was decreased at the edges to preserve the length of the time series. Alternative results with PGFv2 climate forcing and with linear de-trending are shown in Supplementary Figs. 8 and 20, respectively. There is some ambiguity in assigning wheat harvests to calendar years, but our results are robust against shifting the relevant time series by one year (Supplementary Methods).

**Terrestrial ecosystems**. The regions used for analysis are West (7°W–7°E, 42°–50°N) and Central (0°–20°E, 42°–53°N). Unlike in some other sectors, the observational time series for GPP is relatively short—the MODIS data set extends from 2000–2011. For a consistent model-data comparison, the models' GPP anomaly was therefore calculated in terms of the standard deviation over 2000–2010, after removing a linear trend over the period 1979–2010. We note that the standard deviation over this short period, and thus the GPP anomaly metric, may be rather sensitive to inclusion of additional years. Calculating the standard deviation of the models' GPP anomaly over the entire simulated period 1971–2010, rather than just 2000–2010, has little effect on the models' relative distribution, but contracts the spread across models, as can be expected (Supplementary Fig. 21). Note that the conversion of the MODIS space-borne measurements of the fraction of absorbed photosynthetically active radiation (fAPAR) into GPP involves additional uncertainties. To obtain GPP from fAPAR, a light use efficiency (LUE) factor is applied. The MODIS LUE model is based on simple functions of climatic data (i.e., temperature and vapour pressure deficit) calibrated only at a few sites[83]. The fact that the MODIS LUE model ignores soil moisture variations and $CO_2$ physiological effects makes it prone to GPP biases, especially for drought. On the other hand, the negative 2003 summer anomaly found in MODIS GPP is consistent with eddy covariance measurements of carbon fluxes at different sites across Europe[45].

Potential biases in absolute GPP values are also circumvented by our use of standard deviations as unit of comparison.

**Energy**. Hydropower: The anomalies in observed hydropower generation and simulated usable capacity were calculated analogously to the discharge anomaly, as described above; except that instead of a linear trend, a constant mean value was subtracted, since there were no substantial trends. Annual country-level hydro-power generation data was obtained from the EIA database (https://www.eia.gov/beta/international/data/browser/). For some countries, EIA data is limited to the time since their independence; the shortest series (starting 1993) is available for Slovakia and the Czech Republic. Hydropower usable capacity was simulated for the period 1981–2010 based on plant level regulated streamflow simulations from the VIC hydrological model forced with WFDEI climate, hydraulic head, total efficiency of the power generating unit, density of freshwater and the gravitational acceleration (see supplementary information of ref. [56]). Simulations were per-formed for hydropower plants in Europe using data from the World Electric Power Plant Database[84] and simulated usable capacity was then aggregated to country level. The hydropower usable capacity model[56] simulates the physical impacts of changes in water resources on usable power plant capacity. Economic feedbacks of the assessed water constraints and related adaptation options (e.g. on energy prices, supply demands portfolio) are not modelled.

Wind and solar power: In addition to modelling hydropower impacts, we assessed the anomalies in wind power potential capacity factors assuming a homogeneous capacity installed. This assumes the use of a single wind turbine and extrapolation of wind at hub height using the methodology of ref. [85]. We found a record low capacity factor in several Western European countries in summer (June-August) 2003, relative to the 1979–2014 average, using the WFDEI climate data set. In France the anomaly was −39%. While installed wind power capacity was very low in 2003, these results suggest that comparable future climatic events, in a world with much more installed wind power capacity, would induce large negative anomalies in wind power generation. In contrast, we found a positive short-wave radiation anomaly of 5–20% in Western European countries, which would have been relevant for solar power generation. The consequence of all these anomalies would therefore have been a very unusual pattern of energy systems impacts from weather, a situation that transmission operators have to manage in order to balance supply and demand.

**Human health**. Models: We calculate excess mortality attributable to heat for 8 cities for which comparable estimates of observed mortality were found in the literature. Our calculations are based on previously published models called epi-demiological exposure-response functions (ERF). These ERFs were originally derived from a set of non-linear curves developed from a flexible parametric approach[60], and summarised as the percentage change in daily mortality ($b$) per degree increase in maximum daily apparent temperature ($AT_{max}$) above a threshold. Based on a relative risk framework[86], daily excess mortality attributable to heat ($D_{attr}$) can then be derived from the ERFs as follows:

$$D_{attr} = D_{obs}(1 - e^{-b\Delta AT_{max}})$$

where $D_{obs}$ is the observed average daily mortality in the warm season (as reported in ref. [60]). We sum $D_{attr}$ over all days in June to August 2003 on which $AT_{max}$ was above the threshold ($\Delta AT_{max} > 0$). A similar approach has previously been used in climate change impact assessments by refs. [73] and [87]. Here we use the central estimate and the upper and lower 95% confidence intervals of the slope ($b$) reported for each city in ref. [60].

The ERFs used here have been applied for modelling adaptation to heat stress under future climate change[73], but here we assume no adaptation, i.e. the ERFs remain unchanged between the reference period 1990–2000 and 2003. $AT_{max}$ is calculated from daily mean and maximum temperatures and relative humidity, following Tetens' equation[88]. As in ref. [60], we use 3-day averages of $AT_{max}$ in all of our analyses to account for possible lag-effects.

In contrast to the sectors above, excess mortality results are presented not in terms of standard deviations from the long-term average, but in terms of a city's total population. This is because excess mortality is already defined as an excursion from a baseline mortality rate, and is not a continuous variable that could be assumed as approximately normally distributed.

Caveats: We note that the studies we use for evaluating the health models (Supplementary Table 3) differ somewhat in methodology among each other as well as compared to our approach. For instance, while most studies[61,62,87,89–92]—including ours—analyse the whole summer (June–August), two studies[93,94] considered only a shorter period during which the actual heat wave occurred in the respective city. However, since excess mortality occurred mainly during the heat wave days, this difference in approach is expected to have only a minimal effect on the magnitude of the difference between modelled and reported mortality estimates. In addition, our approach only considers the effects of hot temperatures on mortality on each day independent from other days. The models do not account for the potential compounding effects on mortality of heat-wave duration[95,96], nor do they model mortality displacement[97].

Another important caveat is that the ERFs were trained on local weather data obtained from airport weather stations[60], but the outputs from the ERFs are computed by using data from a spatially corresponding 0.5° climate dataset-grid cell as input. To the extent that extreme values may be attenuated at coarser spatial scales, this may bias our results towards lower mortality. We note that this limitation exists in other sectors, too (e.g. when crop models are calibrated on site-level weather variables but forced with grid-level climate data), and for a given location it may be alleviated through bias-correction techniques. However, spatially distributed projections of future climate are typically available on a grid level only, and thus our methodology reflects how the impacts models are typically applied to project future climate change impacts[98,99]. To reiterate, we do not assess how models perform under ideal conditions for a specific event or location, but how well the model and data framework presently available for climate impacts simulations performs for extreme events like the 2003 EHWD.

The ERFs and input climate data do not account specifically for the urban heat island effect, which has been shown to increase local temperature, and in turn, mortality rates[100]. Including this effect would therefore increase our modelled estimates for mortality. However, we do not attempt to include it because the ERFs were trained using local airport weather data and not city-centre weather data[60]. The use of the former for training the ERFs means that the mortality estimates from the ERFs are actually representative of temperatures outside the urban heat island, so it would be inconsistent to add an urban heat island effect to the temperatures that are input to the ERFs. This limitation might explain why our estimates are much lower than the literature values for some cities, as might the potential attenuation of extreme temperatures in the coarse-resolution climate forcing data, as mentioned above. Another potential reason for the discrepancies is that the heat effects on summer mortality in 2003 were probably compounded by high ozone concentrations and resulting poor air quality[18], whereas air quality effects have been removed in the derivation of the ERFs[60].

**Marine ecosystems**. Within ISIMIP2a, an ensemble of global and regional marine ecosystem models was run with forcing from a global coupled ocean circulation and planktonic ecosystem model[101], which in turn was forced with an atmospheric reanalysis dataset[102,103]. This set-up is used as an analogue to the observational forcing of the terrestrial impacts models, given that three-dimensional oceanic observations or reanalysis data are not available. Marine ecosystem models were run without fishing, in order to capture only climate effects on fish biomass. No ocean acidification was assumed.

**Code availability**. Data processing scripts are available from the corresponding author upon request. Availability of impact model code varies by model and is indicated in Supplementary Table 1.

## Data availability

Simulation data from gridded impact models is available through https://esg.pik-potsdam.de/projects/isimip2a/ and citable using the following DOIs: https://doi.org/10.5880/PIK.2017.002 (terrestrial ecosystems); https://doi.org/10.5880/PIK.2017.006 (agriculture); https://doi.org/10.5880/PIK.2017.010 (hydrology); https://doi.org/10.5880/PIK.2018.004 (marine ecosystems, regional); https://doi.org/10.5880/PIK.2018.005 (marine ecosystems, global). The city-level mortality model data and country-level hydropower model data are available from the authors upon request.

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

## Acknowledgements

The work was supported within the framework of the Leibniz Competition (SAW-2013-PIK-5), the EU FP7 project HELIX (grant no. FP7–603864–2), the FP7 project IMPACT2C (grant agreement#282746) and by the German Federal Ministry of Education and Research (BMBF, grant no. 01LS1201A1). H.K. acknowledges support by Grant-in-Aid for specially promoted research 16H06291 from JSPS. Guided access is provided via www.isimip.org. We thank the Global Runoff Data Centre (GRDC) for making available their river discharge data.

## Author contributions

J.Sc. designed the research, analysed and interpreted data, and wrote the paper. S.N.G., C.R., F.Z., P.C., J.E., L.F., V.H., H.K.L., S.I.S., M.T.H.v.V., R.V. and Y.W. helped design the research and interpret results. J.Sc., J.E., K.F., L.W., F.Z., C.R., T.D.E., D.P.T., P.C., H.M.S. and S.N.G. coordinated impact model simulations. J.E., L.F., S.N.G., M.T.H.v.V., R.V., Y.W., F.Z., D.A.C., J.C., M.C., D.D., A.d.W., T.D.E., C.F., D.G., N.H., A.I., N.K., P.L., C.Mo., C.M., H.M.S., S.O., Y.P., T.A.M.P., G.S., Y.S., E.S., T.S., J.Stee., J.Stei., Q.T., H.T. and X.W. performed impact model simulations. L.F. and H.K. provided observational data. M.B. and J.V. processed data. L.B., D.A.C., P.C., M.C., T.D.E., J.E., C.F., L.F., A.D.F., D.G., S.N.G., L.G., N.H., V.H., A.I., N.K., H.K.L., C.Mo., C.Mü., H.M.S., R.O., T.A.M.P., C.R., E.S., S.I.S., T.S., J.Stee., J.Stei., Q.T., H.T., D.P.T., M.T.H.v.V., R.V. and Y.W. provided comments and guidance on the manuscript.

## Additional information

**Competing interests:** The authors declare no competing interests.

Jacob Schewe [1], Simon N. Gosling [2], Christopher Reyer[1], Fang Zhao[3], Philippe Ciais [4], Joshua Elliott[5], Louis Francois[6], Veronika Huber[7], Heike K. Lotze [8], Sonia I. Seneviratne [9], Michelle T.H. van Vliet [10], Robert Vautard[4], Yoshihide Wada [11], Lutz Breuer[12,13], Matthias Büchner[1], David A. Carozza [14,43], Jinfeng Chang [4], Marta Coll [15], Delphine Deryng[16,17], Allard de Wit [18], Tyler D. Eddy [8,19,20], Christian Folberth [11], Katja Frieler[1], Andrew D. Friend[21], Dieter Gerten [1,22], Lukas Gudmundsson [9], Naota Hanasaki [23], Akihiko Ito [23], Nikolay Khabarov [11], Hyungjun Kim [24], Peter Lawrence [25], Catherine Morfopoulos[26], Christoph Müller [1], Hannes Müller Schmied [27,28], René Orth[29,30], Sebastian Ostberg [1], Yadu Pokhrel [31], Thomas A.M. Pugh[32,33], Gen Sakurai[34], Yusuke Satoh[10,23], Erwin Schmid[35], Tobias Stacke[36], Jeroen Steenbeek [37], Jörg Steinkamp [28,44], Qiuhong Tang [38], Hanqin Tian [39], Derek P. Tittensor[8,40], Jan Volkholz[1], Xuhui Wang[4,41,42] & Lila Warszawski[1]

[1]Potsdam Institute for Climate Impact Research, Member of the Leibniz Association, 14473 Potsdam, Germany. [2]School of Geography, University of Nottingham, Nottingham NG7 2RD, UK. [3]School of Geographic Sciences, East China Normal University, Shanghai 200241, China. [4]Laboratoire des Sciences du Climat et de l'Environnement, CEA-CNRS-UVSQ, 91191 Gif-sur-Yvette, France. [5]University of Chicago and ANL Computation Institute, 5735S. Ellis Ave, Chicago, IL 60637, USA. [6]Institut d'Astrophysique et de Géophysique/U.R. SPHERES, Université de Liège, B-4000 LIEGE, Belgium. [7]Department of Physical, Chemical and Natural Systems, Universidad Pablo de Olavide, Ctra. de Utrera 1, 41013 Sevilla, Spain. [8]Department of Biology, Dalhousie University, Halifax, NS B3H 4R2, Canada. [9]ETH Zurich, Land-Climate Dynamics, Institute for Atmospheric and Climate Science, 8092 Zurich, Switzerland. [10]Water Systems and Global Change group, Wageningen University, PO Box 47, 6700 AA Wageningen, The Netherlands. [11]International Institute for Applied Systems Analysis, Schlossplatz 1, A-2361 Laxenburg, Austria. [12]Institute for Landscape Ecology and Resources Management (ILR), Research Centre for BioSystems, Land Use and Nutrition (iFZ), Justus Liebig University Giessen, Heinrich-Buff-Ring 26, 35390 Giessen, Germany. [13]Centre for International Development and Environmental Research (ZEU), Justus Liebig University Giessen, Senckenbergstraße 3, 35392 Giessen, Germany. [14]Department of Earth and Planetary Sciences, McGill University, Montreal H3A 0E8, Canada. [15]Institute of Marine Sciences (ICM - CSIC), Barcelona E-08003, Spain. [16]Leibniz Centre for Agricultural Landscape Research (ZALF), Müncheberg 15374, Germany. [17]IRI THESys, Humboldt University of Berlin, 10117 Berlin, Germany. [18]Wageningen Environmental Research, 6700 AA Wageningen, The Netherlands. [19]Nereus Program, Institute for the Oceans and Fisheries, University of British Columbia, Vancouver V6T 1Z4 BC, Canada. [20]Nereus Program, Institute for Marine & Coastal Sciences, School of the Earth, Ocean, and Environment, University of South Carolina, Columbia 29208 SC, USA. [21]Department of Geography, University of Cambridge, Cambridge CB2 3EN, UK. [22]Geography Department, Humboldt-Universität zu Berlin, 10099 Berlin, Germany. [23]National Institute for Environmental Studies, 16-2 Onogawa, Tsukuba, Ibaraki 305-8506, Japan. [24]Institute of Industrial Science, the University of Tokyo, Tokyo 153-8505, Japan. [25]Terrestrial Science Section, National Center for Atmospheric Research, 1850 Table Mesa Drive, Boulder, CO 80305, USA. [26]Imperial College of London, Department of Life Science, Silwood Park Campus Buckhurst Rd, Berks SL5 7PY, UK. [27]Institute of Physical Geography, Goethe-University Frankfurt, Altenhöferallee 1, 60438 Frankfurt am Main, Germany. [28]Senckenberg Biodiversity and Climate Research Centre (SBiK-F), Senckenberganlage 25, 60325 Frankfurt, Germany. [29]Department of Physical Geography, Bolin Centre for Climate Research, Stockholm University, SE-10691 Stockholm, Sweden. [30]Department of Biogeochemical Integration, Max Planck Institute for Biogeochemistry, D-07745 Jena, Germany. [31]Department of Civil and Environmental Engineering, Michigan State University, MI 48824, USA. [32]School of Geography, Earth & Environmental Sciences, University of Birmingham, Birmingham B15 2TT, UK. [33]Birmingham Institute of Forest Research, University of Birmingham, Birmingham B15 2TT, UK. [34]Institute for Agro-Environmental Sciences, National Agriculture and Food Research Organization, 3-1-3 Kannondai, Tsukuba, Ibaraki 305-8604, Japan. [35]University of Natural Resources and Life Sciences, Vienna, Feistmantelstrasse 4, 1180 Vienna, Austria. [36]Max Planck Institute for Meteorology, 20146 Hamburg, Germany. [37]Ecopath International Initiative, Bellaterra 08193, Spain. [38]Key Laboratory of Water Cycle and Related Land Surface Processes, Institute of Geographic Sciences and Natural Resources Research, Chinese Academy of Sciences, 100101 Beijing, China. [39]School of Forestry and Wildlife Sciences, Auburn University, 602 Duncan Drive, Auburn, AL 36849, USA. [40]UN Environment Programme World Conservation Monitoring Centre, 219 Huntingdon Road, Cambridge CB3 0DP, UK. [41]Sino-French Institute of Earth System Sciences, College of Urban and Environmental Sciences, Peking University, Beijing 100871, China. [42]Laboratoire de Météorologie Dynamique, Université Pierre et Marie Curie, Paris 75005, France. [43]Present address: Department of Mathematics, Université du Québec à Montréal, Montreal H2X 3Y7, Canada. [44]Present address: Johannes Gutenberg-University, Anselm-Franz-von-Bentzel-Weg 12, 55128 Mainz, Germany

