## [Peer Review File · Nature Communications]

Reviewer #1 (Remarks to the Author):

The manuscript explores an important question in climate extremes impacts, and presents results that are intriguing, direction-setting, and highly consequential. The combination of intensifying hazards with growing vulnerabilities and increasing exposure has already resulted in cases of catastrophic multi-sector climate impacts across the globe. The future looks more ominous even as each of the three elements of climate risks (hazards, vulnerability, and exposure) continues to exacerbate (or, in certain cases, continues unabated). The scientific community has begun to realize the significance of the situation. The terminology of "Black Swan" and "Grey Swan" events have started to enter the mainstream scientific literature (e.g., the 2016 paper in *Nature Climate Change* by Ning Lin and Kerry Emanuel on "grey swan" tropical cyclones). Thus, the extent by which multi-sector models of climate impacts may underestimate the damage caused by major climate extremes such as (a future event similar to) the 2003 European Heat Wave and Drought (EHWD), as suggested by this study, is indeed likely to be of immense concern to science and society. This work is timely and deserves to eventually proceed toward publication.

However, the manuscript does have a few glaring problems. First, a critical reader may argue that insights drawn from a sample size of one (i.e., the 2003 EHWD), however massive and wide spread, may not generalize. This may be especially true of the individual impacts on (say) water resources, agriculture, and hydropower, and even in terms of the aggregate impacts. The reader may be dismayed that after putting such a major team and effort together, the authors chose to just look at a single event. As a corollary, the authors may also need to do more due diligence to help the reader understand to what extent their assessment of individual sector-wise impacts for this one event agree with what may already exist in the extant literature. Second, the issue of uncertainty propagation, both from assessments of climate hazards to impacts, and within the impacts models themselves, do not appear to be addressed comprehensively. The careful reader may not get a sense if this is because the uncertainties do not matter so much or whether this is because of the complexity of the uncertainty analysis. Could a full exploration of the uncertainties overwhelm the key insights? Third, certain key questions are left dangling and not fully explored. Thus, the authors make a statement (in the second paragraph) that "the challenges associated with extreme climate events may be larger than the sum of the sectoral impacts" and goes on to provide an example. However, this rather important issue is not really explored in depth or discussed fully in the manuscript. Could this issue be examined as a hypothesis in its own right?

The authors do acknowledge and discuss the problems, in some cases indirectly and/or rather cursorily, but in other cases more directly and/or in relatively greater detail. However, I would be more comfortable if the authors were to reduce the intensity of their assertions through appropriate caveats, spend more time discussing the uncertainties and what these mean in terms of future work, and explain why this paper remains important despite the uncertainties and the caveats because of the kind of questions that are raised and the cautionary alarms that the case study can generate.

I believe this manuscript is an important contribution to the literature and I appreciate the hard work in team building, coordination, and hands-on research that must have gone into the

preparation. However, I also believe that in its current form the manuscript may be open to misinterpretation and exaggeration of claims, unless the uncertainties and caveats are carefully discussed, almost to the extent that would be necessary in a forward-looking perspective type article. On balance, I would recommend publication, but after the concerns have been addressed.

(Signed)

Auroop Ratan Ganguly

Reviewer #2 (Remarks to the Author):

This manuscript explores how multi-model ensemble models for different sectors reproduce the 2003 European heat wave and drought impacts. Sectors include agriculture, fresh water resources, terrestrial and marine ecosystems, energy and human health using a consistent modeling framework. They found that the extreme impacts reported for these sectors is mostly underestimated by most models and by the multi-model ensemble mean and include false alarms in many less-impacted river basins. The authors therefore caution the use of these models for estimating the impact of future extreme event.

This is an interesting, novel and well written manuscript covering a wide range of sectors with broad readership interests.

The conclusions on the performance of the models depends strongly on the quality of the observed data and the setting up of the simulations. For example, river flow data have probably a high quality. But, agricultural commodity reports like from FAO are often based on incomplete farmers reports or are estimates from local experts. The setting of the initial soil water conditions in 2003 (actual values are probably not known for many regions) could also influence the impacts from the low rainfall in the 2003 season on yields. And, the health impacts appear to be widely varying estimates.

It would therefore be good to discuss the conclusions about the model performance also in light of the possible quality of the observed data sources on how well these represent the real impacts.

Minor comments

L54 and, l63, please explain 'low lands'

L287 remove 'very'

Reviewer #3 (Remarks to the Author):

This paper addresses an important issue, namely, how well do available models represent the impacts of climate extremes? The authors use the case of the 2003 European heat wave and drought (EHWD) as a test case and analog for possible future extreme climate events. They make a clear case for using this event for this type of analysis.

Based on the title, I was excited to read this paper. I was expecting to learn how they had implemented a multi-model, multi-sectoral ensemble analysis of a extreme weather event. But as I read the paper, I was disappointed to find that the authors had not done what the title and abstract seemed to be saying. Rather, the paper is reporting on the capability of a set of bio-physical models, independently implemented in response to an historical event, to predict outcomes in each of their respective domains.

So, the first problem I have with this paper is that it mis-represents what is actually being done (or at least it is not clear from the title and the abstract), and leaves out some very important elements (most notably, economic impacts). The phrase “compound impacts” suggests an interaction between the components of the analysis, and the authors argue (page 2) that “challenges associated with extreme climate events may be larger than the sum of the sectoral impacts.” This argument would seem to suggest that the analysis carried out in this study was based on a set of coupled or inter-connect “system of systems” models that would allow this point to be demonstrated. But, alas, this is not what this paper presents. Rather, this paper presents analyses from ensembles of several types of models. What is new here is that each type of model ensemble operates from the same historical weather data to test how well they predicted their respective impact indicators over the EHWD event. But the models do not interact the way an integrated assessment system-of-systems analysis would do.

The models in this study are: water resources, agriculture, terrestrial ecosystems, energy, human health and marine ecosystems. The analysis shows that they individually tend to underestimate the impacts of the EHWD. The authors claim this is the first time such an exercise has been carried out (but one would have to qualify this claim, this may be true for this many types of models, but other multi-sectoral analyses of extreme events have been done in the voluminous literature on natural disasters, and many of those are using coupled models). I find the resulting analysis to be useful, and agree that this finding shows the need for more analysis of extreme events (but we already knew that).

A major missing element of the set of models in this study is economic models, and this in turn explains the lack of economic analysis. Economic analysis is one way that the physical impacts could be aggregated, and it would also provide a meaningful way to evaluate the distribution of impacts on the most vulnerable elements of society. This distributional or vulnerability dimension is an obvious missing element (except for the health part of the analysis where you could say that vulnerability is taken into account implicitly). At a minimum, the authors should have pointed out

this important needed extension of what they are doing, e.g., by linking their physical model analyses to an economic impact assessment. They will probably respond to this criticism by saying that they cannot do everything, and that such an analysis is beyond the scope of what they could do. But that is not a justification for ignoring the importance of economic impacts and not explaining why their “impacts” indicators may have limited value. Nor is this a justification for presenting their work as if it represented “compound” impacts when it does not.

The lack of any economists on this team also is telling in the discussion. For example, the paper reads as if physical impacts, e.g., on crop yields, is the relevant measure of impact on agriculture or society. But this is clearly not true, e.g., crop prices typically rise during droughts, so farmers’ income losses are offset by higher prices, and may be offset by subsidies or insurance, but consumers (especially poor ones) are likely to be more adversely impacted. This is the type of information that could be provided by a more complete analysis that included economic impacts. Additionally, if the water models were linked to the crop and economic models, the potential “compound” impacts of the extreme event, including the impacts on public finance that the authors allude to, could be quantified. The authors refer to “compound” effects of extreme events, such as multiple pressures on public finances, but do not include any economic analysis that could represent such effects.

I am also surprised by the authors’ health analysis, the model is extremely simplistic. I am not an expert in that field, but I surprised that there is nothing better than a simple exponential relationship between temperature and mortality?

In summary, there is some useful research here about the comparative capability of modeling to represent the effects of extremes in important physical dimensions. However, as I noted at the outset, I find the paper is substantially over-selling what is being done – the authors are not simulating the “compound” impacts of climate extremes. Rather, the paper is reporting on the capability of a set of bio-physical models to predict a set of impact indicators in response to an historical extreme event. Thus, I conclude the paper should not be published. That being said, I think the material in this paper does deserve eventual publication in some form, appropriately presented.

Point-by-point list of reviewer comments (black) and responses (blue)

Reviewer 1

“The manuscript explores an important question in climate extremes impacts, and presents results that are intriguing, direction-setting, and highly consequential. The combination of intensifying hazards with growing vulnerabilities and increasing exposure has already resulted in cases of catastrophic multi-sector climate impacts across the globe. The future looks more ominous even as each of the three elements of climate risks (hazards, vulnerability, and exposure) continues to exacerbate (or, in certain cases, continues unabated). The scientific community has begun to realize the significance of the situation. The terminology of "Black Swan" and "Grey Swan" events have started to enter the mainstream scientific literature (e.g., the 2016 paper in Nature Climate Change by Ning Lin and Kerry Emanuel on "grey swan" tropical cyclones). Thus, the extent by which multi-sector models of climate impacts may underestimate the damage caused by major climate extremes such as (a future event similar to) the 2003 European Heat Wave and Drought (EHWD), as suggested by this study, is indeed likely to be of immense concern to science and society. This work is timely and deserves to eventually proceed toward publication.”

We thank the reviewer for reading our manuscript and we are pleased that they consider our research to be timely, important and direction-setting.

“First, a critical reader may argue that insights drawn from a sample size of one (i.e., the 2003 EHWD), however massive and wide spread, may not generalize. This may be especially true of the individual impacts on (say) water resources, agriculture, and hydropower, and even in terms of the aggregate impacts. The reader may be dismayed that after putting such a major team and effort together, the authors chose to just look at a single event. As a corollary, the authors may also need to do more due diligence to help the reader understand to what extent their assessment of individual sector-wise impacts for this one event agree with what may already exist in the extant literature.”

Although reviewer 3 notes that we “make a clear case for using this event for this type of analysis” we agree with Reviewer 1 that we could have discussed the rationale for focusing on a single event in more detail. Our motivation for the choice of the 2003 EHWD is now stated clearly in the Introduction of the revised manuscript, so that it is clear up-front why we chose this one event:

“...these complex impact models, in conjunction with global climate models, form the basis of much of our current knowledge about future global climate change impacts, as reflected in the Intergovernmental Panel on Climate Change reports, for instance. Whether they capture extreme events well is therefore a key concern even beyond the application in economic assessments.

And yet, it is not known how well the current suite of models can reproduce the multi-sectoral impacts of a given climatic extreme event. Global process-based impact models have been evaluated in terms of average quantities and sometimes in terms of inter-annual or intra-annual variability (Beck et al., 2016; Hattermann et al., 2017; Müller et al., 2017; Traore et al., 2014), but

their performance under extreme conditions has rarely been tested at large spatial scale (Schauberger et al., 2017), and never – to our knowledge – in a multi-sector setting. And since events that are very rare today may become much more frequent in the future (Dosio, Mentaschi, Fischer, & Wyser, 2018), testing for variability alone may not be enough.

Here, we choose the 2003 European heat wave and drought (EHWD) event as a test case. The EHWD was substantially stronger than previously observed events; it severely impacted several important sectors across a large geographical area, and its impacts are relatively well documented.”

We then discuss the EHWD and its impacts in detail, including its role as a harbinger of future extreme events, in the Results section. In the Discussion section, we discuss the implications of our study beyond the particular event of 2003:

“We have investigated a single climate event, but our results are relevant beyond the case of the 2003 EHWD. In 2015 and 2018, central Europe was again hit by extremely hot and dry summer conditions, and projected trends in temperature and precipitation suggest that such events will occur more frequently in the future (Orth et al., 2016). While each event is unique, similar events produce similar biophysical impacts: future drought-heat wave events will again lead to low river flows, reduce ecosystem productivity, and damage crop yields. As long as the model shortcomings exposed by our analysis persist, they will likely lead to biased estimates of extreme event impacts.

This is followed by two paragraphs summarizing the different types of model shortcomings.

To address the reviewer’s final point above, which is about how our assessment compares with what already exists in the literature, we note that there are very few comparable model evaluation studies. As we state in the Introduction,

“Global process-based impact models have been evaluated in terms of average quantities and sometimes in terms of inter-annual or intra-annual variability (Beck et al., 2016; Hattermann et al., 2017; Müller et al., 2017; Traore et al., 2014), but their performance under extreme conditions has rarely been tested at large spatial scale (Schauberger et al., 2017)...”

In the revised manuscript, we have extended our discussion of how our results relate to the above and other model evaluation studies in the different sectors. For water resources:

“Our results are consistent with recent studies attesting a relatively good performance of global hydrological models for discharge and runoff variability in European catchments, compared to other regions (Beck et al., 2016; Zaherpour et al., 2018). At the same time, the tendency towards false-alarms is in agreement with a dry bias induced by the models’ potential evaporation schemes (Milly & Dunne, 2017). The fact that one of our models (number 6) appears consistently at the dry end of the ensemble may be because it applies a temperature-based evaporation scheme (Hamon) that has been shown to induce a particularly large bias when applied outside its calibration range (Milly & Dunne, 2017).”

For agriculture:

“The best-matching models differ from country to country, in line with a previous evaluation of these models (Müller et al., 2017) which also found mixed skill in reproducing overall inter-annual yield variability.”

“The crop models assume full irrigation in irrigated areas and do not account for potential limitations in water availability due to drought, which induces an overestimation of irrigated yields, and thus biases total yields in countries with much irrigation, such as France or Italy (Supplementary Figure S9). This is in line with a recent study showing that extreme heat leads to strong declines in maize yield in both observations and models only under rainfed conditions (Schauberger et al., 2017).”

For GPP:

“Our finding of a smaller sensitivity of GPP to drought in models than in observations is consistent with a previous multi-model study which, however, assessed a shorter time span (Reichstein et al., 2007). Inspecting individual models’ GPP time series (Supplementary Figures S11 and S12), we note that those models that most closely reproduce the 2003 relative anomaly also exhibit a pronounced positive anomaly in 2007 and 2008, consistent with the MODIS estimates. The large spread across models for the reduction of GPP in the 2003 EHWI is despite a relatively high spatial correlation with MODIS-derived GPP globally in all these models (Ito et al., 2017).”

For hydropower:

“A previous evaluation of the hydropower model, using the same climate input, found relatively low root mean squared error in southern European countries, but larger values in some north European countries, with particularly poor performance in Latvia (Michelle T. H. van Vliet, Wiberg, Leduc, & Riahi, 2016). Our analysis broadly confirms this regional pattern, while focusing explicitly on a drought event.”

For human health:

“To our knowledge, this is the first independent evaluation of the performance of these models outside their calibration period.”

“Second, the issue of uncertainty propagation, both from assessments of climate hazards to impacts, and within the impacts models themselves, do not appear to be addressed comprehensively. The careful reader may not get a sense if this is because the uncertainties do not matter so much or whether this is because of the complexity of the uncertainty analysis. Could a full exploration of the uncertainties overwhelm the key insights?”

We agree that uncertainties at different points in the modeling chain need to be carefully considered. Indeed, this is a main motivation for our modeling approach: by considering multiple models in each sector, and multiple climate forcing products to drive these models, we are able to map the important dimensions of uncertainty mentioned by the reviewer – the reconstruction of historical atmospheric

fields from weather observations, and the structure and parameterization of the biophysical impact models – to a large extent.

We have extended our analysis of the three different climate forcing products, including maps and time series of both temperature and precipitation anomalies for each data set in Supplementary Figures S1 and S2. In the Methods section, we state that:

“The magnitude and spatio-temporal pattern of the 2003 summer temperature and precipitation anomaly is similar in all three data sets (Supplementary Figures S1 and S2), and in agreement with other reanalysis products (García-Herrera, Díaz, Trigo, Luterbacher, & Fischer, 2010; Schär et al., 2004).”

Accordingly, forcing uncertainty plays a minor role, and we now clarify this in the Discussion & Conclusions section:

“We also find that generally the choice of climate forcing data set is not a major source of uncertainty.”

Model structural uncertainty is more important, and we discuss its implications, and relevance for integrated assessments, in the Discussion & Conclusions section:

“Our results emphasize the importance of considering the uncertainties of current biophysical impact estimates. Modelling studies of future climate change impacts on mean conditions in different sectors (Caminade et al., 2014; Friend et al., 2013; Rosenzweig et al., 2014; Schewe et al., 2014) have shown that the uncertainties of biophysical impact estimates are of a similar magnitude as climate modelling uncertainty. A common practice has since been to use multi-model ensembles rather than single impact models, as is standard in climate modelling. We have shown that for an extreme event, even an entire ensemble of impact models can sometimes completely miss observations. This urgently calls for a shift in the agenda of model development and evaluation away from mean conditions towards extremes. In the meantime, a precautionary interpretation of impact estimates is recommended, where the most pessimistic model in an ensemble must be considered just as plausible as those in the centre of the distribution.”

“Third, certain key questions are left dangling and not fully explored. Thus, the authors make a statement (in the second paragraph) that “the challenges associated with extreme climate events may be larger than the sum of the sectoral impacts” and goes on to provide an example. However, this rather important issue is not really explored in depth or discussed fully in the manuscript. Could this issue be examined as a hypothesis in its own right?”

We have completely revised the Introduction in response to the three reviewers’ comments. The above statement has been removed in order to lend more focus to the Introduction. However, the interactions between sectors are an important motivation for studying impacts in different sectors together in a consistent framework. In the revised manuscript, we cover this topic in the Discussion & Conclusions section, and illustrate the example of water resource constraints on crop yields using our simulation results:

“...In process-based models, the capacity to respond to extreme climate conditions can be overestimated if resource constraints are not accounted for. An example is the lack of constraints on irrigation water in our crop model simulations. For instance, streamflow in the Vienne basin, which contains large shares of irrigated maize cropland (van der Velde, Wriedt, & Bouraoui, 2010), was anomalously low in summer 2003 (Figure 2); and indeed, the maize yield anomaly in France is simulated more realistically when irrigation is switched off, than in the default model setup with irrigation (Supplementary Figure S9). Accounting for such cross-sectoral interactions (Frieler et al., 2015) and reflecting changing levels of societal adaptation in models will be critical for more accurate impact and damage estimates in the future.”

“The authors do acknowledge and discuss the problems, in some cases indirectly and/or rather cursorily, but in other cases more directly and/or in relatively greater detail. However, I would be more comfortable if the authors were to reduce the intensity of their assertions through appropriate caveats, spend more time discussing the uncertainties and what these mean in terms of future work, and explain why this paper remains important despite the uncertainties and the caveats because of the kind of questions that are raised and the cautionary alarms that the case study can generate.”

This is a very useful suggestion, and the revised manuscript reflects our efforts to improve on these points. For example, we have reduced the intensity of our assertions already in the abstract, by removing the potentially misleading reference to “compound impacts”, and by lending more nuance to our quantitative statements:

“...we find that a majority of models underestimate the extremeness of impacts in important sectors ...”

In the manuscript, we are clear about the general caveats to our research:

“Important natural processes such as heat stress effects on ecosystem photosynthesis are also missing in some models, while other processes are parameterised through empirical relationships that may not hold outside past ranges of climatic variability (Milly & Dunne, 2017). Such shortcomings will affect impact assessments of other climatic events, too. In addition, there are known issues that have not emerged in our study, but may prove problematic when assessing the impacts of extreme events of different types, or outside Europe. For instance, the global ecosystem models do not consider the potentially amplifying interactions between climate change and forest disturbances, such as wind storms (Reyer et al., 2017); and some global hydrological models struggle to capture the magnitude and timing of processes such as transmission loss and snowmelt accurately (Zaherpour et al., 2018). Finally, our models may perform worse in parts of the developing world where constraints from observational data are more limited.”

We are also clear about some of the more specific caveats for individual sectors, e.g. for agriculture:

“In process-based models, the capacity to respond to extreme climate conditions can be overestimated if resource constraints are not accounted for. An example is the lack of constraints on irrigation water in our crop model simulations. For instance, streamflow in the Vienne basin,

which contains large shares of irrigated maize cropland (van der Velde et al., 2010), was anomalously low in summer 2003 (Figure 2); and indeed, the maize yield anomaly in France is simulated more realistically when irrigation is switched off, than in the default model setup with irrigation (Supplementary Figure S8). Accounting for such cross-sectoral interactions (Frieler et al., 2015) and reflecting changing levels of societal adaptation in models will be critical for more accurate impact and damage estimates in the future.”

And also for health (in the Supplementary Material):

“We note that the studies we use for evaluating the health models (Table S3) differ somewhat in methodology among each other as well as compared to our approach. For instance, while most studies (Borrell et al., 2006; Grize, Huss, Thommen, Schindler, & Braun-Fahrländer, 2005; Martínez Navarro, Simón-Soria, & López-Abente, 2004; P Michelozzi et al., 2005; Paola Michelozzi et al., 2006; Mitchell et al., 2016; Tobías et al., 2010) – including ours – analyze the whole summer (June-August), two studies (Johnson et al., 2005; Le Tertre et al., 2006) considered only a shorter period during which the actual heat wave occurred in the respective city. However, since excess mortality occurred mainly during the heat wave days, this difference in approach is expected to have only a minimal effect on the magnitude of the difference between modelled and reported mortality estimates. In addition, our approach only considers the effects of hot temperatures on mortality on each day independent from other days. The models do not account for the potential compounding effects on mortality of heat-wave duration (Gasparrini & Armstrong, 2011; Hajat et al., 2006), nor do they model mortality displacement (Gosling, Lowe, McGregor, Pelling, & Malamud, 2009).”

We also explain what the uncertainties, and our conclusions in general, mean in terms of future work:

“Modelling studies of future climate change impacts on mean conditions in different sectors (Caminade et al., 2014; Friend et al., 2013; Rosenzweig et al., 2014; Schewe et al., 2014) have shown that the uncertainties of biophysical impact estimates are of a similar magnitude as climate modelling uncertainty. A common practice has since been to use multi-model ensembles rather than single impact models, as is standard in climate modelling. We have shown that for an extreme event, even an entire ensemble of impact models can sometimes completely miss observations. This urgently calls for a shift in the agenda of model development and evaluation away from mean conditions towards extremes. In the meantime, a precautionary interpretation of impact estimates is recommended, where the most pessimistic model in an ensemble must be considered just as plausible as those in the centre of the distribution.

For quantifying the ultimate impacts of climate change on different aspects of the economy, or parts of society, different metrics need to be applied beyond the (mainly) biophysical indicators studied here; e.g., macro-economic damages, public health burdens, or effects on societal equality and political stability. However, any such analysis needs to rest on a firm assessment of the biophysical impact, and will be unreliable if the biophysical impact is not captured accurately. Our study thus provides important context to any subsequent analysis of economic and societal impacts.”

Reviewer 2

“This is an interesting, novel and well written manuscript covering a wide range of sectors with broad readership interests. The conclusions on the performance of the models depends strongly on the quality of the observed data and the setting up of the simulations. For example, river flow data have probably a high quality. But, agricultural commodity reports like from FAO are often based on incomplete farmers reports or are estimates from local experts. The setting of the initial soil water conditions in 2003 (actual values are probably not known for many regions) could also influence the impacts from the low rainfall in the 2003 season on yields. And, the health impacts appear to be widely varying estimates.

It would therefore be good to discuss the conclusions about the model performance also in light of the possible quality of the observed data sources on how well these represent the real impacts.”

We thank the reviewer for their positive comments on the broad interest and novelty of our manuscript.

Even though the 2003 EHWD is one of the extreme climate events that is studied the most, the observed data available for the event has its limitations. This is an important caveat and we agree that it deserves more comprehensive analysis and discussion in the manuscript. To this end, we have complemented the analysis with additional data sources, where they are of sufficient quality and are available, and we have also engaged in a more comprehensive discussion of the issue of observed data quality. We agree with the reviewer’s comment on the quality of the river flow data but taking into account their comments on the other sectors we have paid particular attention to them. We summarise these additions to the manuscript below.

For agriculture, the FAOSTAT database, while comprehensive and consistent, may not always be accurate. Estimates of the 2003 EHWD’s effect on crop yields are also available from COPA-COGECA. To this end, we have included these in the revised manuscript, in order to complement the FAOSTAT values:

“Consistent with previous studies (Ciais et al., 2005) and early assessments by the COPA-COGECA agricultural association (Copa-Cogeca, 2004; Supplementary Table S2), we find large negative yield anomalies in the FAOSTAT data set in France, Germany and Italy for both maize and wheat; as well as in Spain for maize, and in Austria and Portugal for wheat (Figure 3, black circles).”

Furthermore, in the new Supplementary Table S2 we compare yield changes reported by FAOSTAT and COPA-COGECA for the countries covered in the COPA-COGECA report; they largely agree except for maize yields in Austria, which we discuss:

“We also note that COPA-COGECA report a decline in maize yields in Austria by about 10% between 2002 and 2003, which is not reported in the FAOSTAT data; thus, the real observational value in Figure 3 may be closer to the model ensemble mean than the FAOSTAT value shown by the black circle (Supplementary Table S2).”

For temperature-related mortality, the existence of multiple independent studies for some cities allows one to assess the range of values in which the true value might lie. Sometimes the ranges are large, e.g.

in the case of Barcelona, estimates vary between 26 and 42 excess deaths per 100,000, but one can still judge whether the model results are in the same order of magnitude as observations. Moreover, there are many uncertainties in estimating observed heat-related mortality, not least the way in which a heat-related death is defined (excess deaths can be computed from running means of varying length). To some extent, therefore, having a range of observed values to compare simulations to, is advantageous over having a single value, because the range removes any misguided assumptions on the precision and accuracy of the observed data.

For marine ecosystems, we have used reported fish catch data which have only recently become available through the Sea Around Us project, and which are the highest quality data available.

We have also used the best observational data sources known and available to us, for terrestrial ecosystems and energy. However, we acknowledge that they carry substantial uncertainties, which were discussed in the original manuscript. In the revised manuscript, we have added the following discussion concerning MODIS data, to the terrestrial ecosystems Methods and Supplementary Material, in order to provide further context as regards the reviewer's points above:

“Note that the conversion of the MODIS space-borne measurements of absorbed photosynthetically active radiation into GPP involves additional uncertainties (Supplementary Material). While this means that the MODIS GPP estimates may be prone to errors, the negative 2003 summer anomaly found in MODIS GPP is consistent with eddy covariance measurements of carbon fluxes at different sites across Europe (Reichstein et al., 2007). Potential biases in absolute GPP values are also circumvented by our use of standard deviations as unit of comparison.”

“We use MODIS GPP estimates as an observational reference data set, because they are based on satellite data and cover a large geographic area and relatively long time span. However, the variable that is retrieved from satellites is not GPP, but fAPAR, the fraction of absorbed photosynthetically active radiation. To obtain GPP from PAR and fAPAR, a light use efficiency (LUE) factor is needed, which comes from a model. The MODIS LUE model is based on simple functions of climatic data (i.e., temperature and vapour pressure deficit) calibrated only at few sites (Running & Zhao, 2015). The lack of soil moisture dependence of the MODIS LUE model as well as ignored CO₂ physiological effects makes it prone to GPP biases, especially for drought. On the other hand, the negative 2003 summer anomaly found in MODIS GPP is consistent with eddy covariance measurements of carbon fluxes at different sites across Europe (Reichstein et al., 2007).”

As mentioned in response to Reviewer 1, we have also added further analysis of the different climate forcing datasets, including additional figures showing the differences between the datasets for the most relevant variables temperature and precipitation, and discussing the implications of these differences for the impact estimates.

Reviewer 3

“This paper addresses an important issue, namely, how well do available models represent the impacts of climate extremes? The authors use the case of the 2003 European heat wave and drought (EHWD) as a test case and analog for possible future extreme climate events. They make a clear case for using this event for this type of analysis.”

We welcome the reviewer’s comments on the importance of the issues that we address and our rationale for selecting a single event for the analysis.

“...the first problem I have with this paper is that it mis-represents what is actually being done (or at least it is not clear from the title and the abstract), and leaves out some very important elements (most notably, economic impacts). The phrase “compound impacts” suggests an interaction between the components of the analysis, and the authors argue (page 2) that “challenges associated with extreme climate events may be larger than the sum of the sectoral impacts.”

We used the term “compound impacts” to refer to the co-occurrence of different biophysical impacts of an extreme climate event. “Coincidental” may have been a better word, upon reflection. We used the term in the original submission because in our modeling framework the impacts are analysed in a consistent way and can therefore be “compounded”. The same word has been used elsewhere (see Leonard et al., 2014, for a broad definition of compound events). Nevertheless, we recognise that this language may be prone to misinterpretation, especially in a journal with a wide readership, as the term can mean different things to researchers working in different disciplines. To this end we have removed the term “compound impacts” from the title, abstract, and manuscript body.

We have carefully considered the reviewer’s comment about missing an economic analysis from our assessment. There are several reasons why an economic analysis is not included in our study. Cross-sectoral economic assessments of climate change impacts mostly rely on integrated assessment models. Such models are based on simplified representations of the climate system and the biophysical impacts that arise from them (e.g. in the form of statistical damage functions), and they consider long-term equilibrium responses. This is sometimes referred to as the “top-down” approach to climate change economic impact assessment. An important step forwards in how economic impacts are estimated was made in a recent study in *Nature Communications* (Monier et al., 2018). The paper outlines how through a more bottom-up approach to assessment, more complex physical models can be integrated with an economic model to assess cross-sectoral linkages. However, Monier et al., (2018) only include single models of sectoral impacts and, importantly, only considers changes in average conditions on those sectors. Their study therefore ignores the impacts of extreme events and impact model uncertainty. A novelty of our analysis, therefore, is that we provide a unique context to these important issues through a *multi-model* multi-sector analysis that rigorously accounts for modeling uncertainty and acknowledges the importance of *extreme events*. We intend to do more than replicate Monier et al. (2018) with multiple impact models – rather we seek to provide a context to such studies by demonstrating the ability of, and inherent uncertainty in, complex impact models to represent an extreme event.

We believe that the impacts of extreme events are an important yet understudied component of the total damages caused by past and future climate change. If they are to be represented in economic assessments of climate change, then a major prerequisite is a thorough understanding of our ability to

simulate such extreme impacts in the first place. This is the goal of our study: it brings together the state-of-the-art biophysical models across sectors, and evaluates them using a well-studied past climate extreme of a type that will occur more frequently in the future. This has never been done before. It means that we can provide an important direction-setting message (as noted by Reviewer 1), which is to caution against economic assessments unless the representation of extreme events is improved and the impact model uncertainty is accounted for; otherwise such assessments may underestimate the actual damages substantially:

“Our results emphasize the importance of considering the uncertainties of current biophysical impact estimates. Modelling studies of future climate change impacts on mean conditions in different sectors (Caminade et al., 2014; Friend et al., 2013; Rosenzweig et al., 2014; Schewe et al., 2014) have shown that the uncertainties of biophysical impact estimates are of a similar magnitude as climate modelling uncertainty. A common practice has since been to use multi-model ensembles rather than single impact models, as is standard in climate modelling. We have shown that for an extreme event, even an entire ensemble of impact models can sometimes completely miss observations. This urgently calls for a shift in the agenda of model development and evaluation away from mean conditions towards extremes. In the meantime, a precautionary interpretation of impact estimates is recommended, where the most pessimistic model in an ensemble must be considered just as plausible as those in the centre of the distribution”

Moreover, we now acknowledge in our concluding remarks the implications of our study for economic impact assessments:

“For quantifying the ultimate impacts of climate change on different aspects of the economy, or parts of society, different metrics need to be applied beyond the (mainly) biophysical indicators studied here; e.g., macro-economic damages, public health burdens, or effects on societal equality and political stability. However, any such analysis needs to rest on a firm assessment of the biophysical impact, and will be unreliable if the biophysical impact is not captured accurately. Our study thus provides important context to any subsequent analysis of economic and societal impacts”

We regret that our aim and motivation was not laid out well in our original manuscript, and apologise for any confusion arising from that. We have completely revised the Title, Abstract, Introduction, and large parts of the Discussion & Conclusions section in order to clarify what the paper does and what it does not, and why. For instance, in the Introduction:

“Estimation of the total damages caused by climate change requires a quantification of climate impacts across a large range of economic and societal sectors. These sectors include agriculture (Rosenzweig et al., 2014), water resources (Schewe et al., 2014), energy supply and demand (M.T.H. van Vliet et al., 2016), human health (Whitmee et al., 2015), and ecosystem services (Pecl et al., 2017). There are approaches that integrate damages across sectors, such as the highly idealised damage functions used in integrated assessment modelling (Pindyck, 2013), but also more sophisticated, coupled economic modelling frameworks that combine individual sectoral models (Harrison et al., 2018; Hsiang et al., 2017; Monier et al., 2018). However, these

approaches are centred on gradual changes in physical and biophysical indicators – such as crop yields or water resources – and largely ignore the impacts of extreme climate and weather events.

This is a serious research gap because such events cause enormous damages (Franzke, 2017). For addressing it, sectoral impact models must be able to credibly represent the impacts of extreme events. The goal of this paper is to test whether this is the case in the complex, process-based impact models that are routinely being applied in global-scale climate impact assessment (Rosenzweig et al., 2014; Schewe et al., 2014; M.T.H. van Vliet et al., 2016). While these models may be too costly to integrate them directly in cross-sectoral economic models, they are the benchmark for any simpler models. More generally, these complex impact models, in conjunction with global climate models, form the basis of much of our current knowledge about future global climate change impacts, as reflected in the Intergovernmental Panel on Climate Change reports, for instance. Whether they capture extreme events well is therefore a key concern even beyond the application in economic assessments.

And yet, it is not known how well the current suite of models can reproduce the multi-sectoral impacts of a given climatic extreme event. Global process-based impact models have been evaluated in terms of average quantities and sometimes in terms of inter-annual or intra-annual variability (Beck et al., 2016; Hattermann et al., 2017; Müller et al., 2017; Traore et al., 2014), but their performance under extreme conditions has rarely been tested at large spatial scale (Schauberger et al., 2017), and never – to our knowledge – in a multi-sector setting. And since events that are very rare today may become much more frequent in the future (Dosio et al., 2018), testing for variability alone may not be enough.

Here, we choose the 2003 European heat wave and drought (EHWD) event as a test case. The EHWD was substantially stronger than previously observed events; it severely impacted several important sectors across a large geographical area, and its impacts are relatively well documented. We examine the impacts of the EHWD in a large ensemble of state-of-the-art impact models covering agriculture, water resources, terrestrial and marine ecosystems, energy, and human health, for the first time in a common modelling framework. For each of these sectors, we identify key observed impacts of the 2003 EHWD reported in the literature and/or recorded in public databases, and examine how closely the models – driven by observations-based climate data – reproduce those impacts. As a common impact metric, we choose the deviation of 2003 from the historical average, adjusted for long-term trends, and normalised by the historical standard deviation (except for human health; see Methods). We thereby circumvent potential biases in the baseline or the average inter-annual variability, and instead focus on the models' ability to pick out the anomalous 2003 event from the rest of the time series.

We find that generally the spread between models within a sector is large; neither the multi-model mean nor any individual model reliably reproduce observations; and models more often underestimate than overestimate the extremeness of observed impacts. We discuss potential reasons for these findings and their implications for integrated assessments of climate change impacts and for future model development.”

“What is new here is that each type of model ensemble operates from the same historical weather data to test how well they predicted their respective impact indicators over the EHWD event. But the models do not interact the way an integrated assessment system-of-systems analysis would do.”

Whilst in response to the reviewer’s previous comment we noted the novelties of our analysis, the reviewer correctly identifies here one of the limitations of our approach. However, integration of impact models was not the goal of our study. Instead, we set out to assess the ability of the biophysical models in different sectors to estimate the biophysical impacts of the 2003 EHWD. As mentioned above, recent integrated assessments apply, at best, a single biophysical model in each impact sector, often at coarser spatial and temporal resolution than our models, and ignoring the impacts of extreme events (Harrison et al., 2018; Monier et al., 2018). On the other hand, efforts to dynamically couple multiple biophysical impact models with a physical climate model have so far resulted in models (such as e.g. the CESM, Hurrell et al., 2013) which, though integrated, do not resolve some of the sectors discussed here at all, and other sectors at much less detail than our sectoral models do.

Our study does not replace such integrated studies, but provides important context to them, by i) highlighting the importance of extreme events for estimating the overall damages from climate change; ii) demonstrating that the estimates of biophysical impacts of an extreme event in several important sectors can differ substantially between state-of-the-art models, even when using the same climate forcing, and are often much less extreme than the observed impacts; iii) discussing reasons for these deviations, and iv) discussing the implications of these findings for the representation of biophysical impacts in integrated assessments, as well as outlining some important considerations for improving such assessments. We recognize that the aims of our study were not laid out clearly in our original manuscript, and that points iii) and iv) in particular were not given sufficient treatment. We have addressed these shortcomings in the revised manuscript.

“The models in this study are: water resources, agriculture, terrestrial ecosystems, energy, human health and marine ecosystems. The analysis shows that they individually tend to underestimate the impacts of the EHWD. The authors claim this is the first time such an exercise has been carried out (but one would have to qualify this claim, this may be true for this many types of models, but other multi-sectoral analyses of extreme events have been done in the voluminous literature on natural disasters, and many of those are using coupled models). I find the resulting analysis to be useful, and agree that this finding shows the need for more analysis of extreme events (but we already knew that).”

We are glad that the reviewer finds the analysis useful. To our knowledge this is the first study that uses past observational data to evaluate impact models’ performance for an extreme event across different sectors. It thus goes beyond existing literature on the impacts of extreme events in general in individual sectors (e.g. Frank et al., 2015), or that on projected future climate change impacts across sectors (e.g. Arnell et al., 2013; Piontek et al., 2014), or that on statistical evaluation of impact models in individual sectors (e.g. Huang et al., 2017, and other studies cited in our manuscript).

“A major missing element of the set of models in this study is economic models, and this in turn explains the lack of economic analysis. Economic analysis is one way that the physical impacts could be aggregated, and it would also provide a meaningful way to evaluate the distribution of impacts on the

most vulnerable elements of society. This distributional or vulnerability dimension is an obvious missing element (except for the health part of the analysis where you could say that vulnerability is taken into account implicitly). At a minimum, the authors should have pointed out this important needed extension of what they are doing, e.g., by linking their physical model analyses to an economic impact assessment. They will probably respond to this criticism by saying that they cannot do everything, and that such an analysis is beyond the scope of what they could do. But that is not a justification for ignoring the importance of economic impacts and not explaining why their “impacts” indicators may have limited value. Nor is this a justification for presenting their work as if it represented “compound” impacts when it does not.”

We have carefully considered the reviewer’s opinion on our lack of economic analysis. We noted earlier in our response to the reviewer that our use of the word “compound” was somewhat ambiguous (and thus it is removed from the revised manuscript). The original use of this term inadvertently implied a degree of aggregation of impacts and we therefore understand why the reviewer notes that an economic analysis is one way that the physical impacts could be aggregated. We agree that such an approach would be worthwhile if impacts were to be aggregated, but the aggregation of impacts is not a goal of our study. Rather it is to demonstrate the ability of *physical* models to estimate *physical* impacts, not *economic* impacts. To further address this comment, we clarify in the revised manuscript the role and importance of our analysis for economic assessments of climate impacts, in the Introduction – as cited above in response to this reviewer’s comment – as well as in the Discussion section, as follows:

“For quantifying the ultimate impacts of climate change on different aspects of the economy, or parts of society, different metrics need to be applied beyond the (mainly) biophysical indicators studied here; e.g., macro-economic damages, public health burdens, or effects on societal equality and political stability. However, any such analysis needs to rest on a firm assessment of the biophysical impact, and will be unreliable if the biophysical impact is not captured accurately. Our study thus provides important context to any subsequent analysis of economic and societal impacts.”

The lack of any economists on this team also is telling in the discussion. For example, the paper reads as if physical impacts, e.g., on crop yields, is the relevant measure of impact on agriculture or society. But this is clearly not true, e.g., crop prices typically rise during droughts, so farmers’ income losses are offset by higher prices, and may be offset by subsidies or insurance, but consumers (especially poor ones) are likely to be more adversely impacted. This is the type of information that could be provided by a more complete analysis that included economic impacts. Additionally, if the water models were linked to the crop and economic models, the potential “compound” impacts of the extreme event, including the impacts on public finance that the authors allude to, could be quantified. The authors refer to “compound” effects of extreme events, such as multiple pressures on public finances, but do not include any economic analysis that could represent such effects.

We agree that different metrics need to be applied to measure the ultimate impacts on different aspects of the economy, or parts of society. However, as we mention in the revised manuscript, any such analysis needs to rest on a firm assessment of the physical impact. If the physical impact is not captured accurately, any estimate of the societal impacts will be unreliable, regardless of the metric applied. Our

study thus provides important context to any subsequent analysis of economic and societal impacts, because it is, to our knowledge, the first study that uses past observational data to evaluate impact models' performance for an extreme event across different sectors. We have revised the manuscript extensively in order to clarify this.

We share the reviewer's aspiration for linking impact models but this has to be tempered with the acknowledgement that fully coupled cross-sectoral simulations, while desirable, are still quite a way off from being realised. The ISIMIP ensemble used here is a very recent development and the only ensemble of complex biophysical impact models that has been designed such that simulations are consistent and comparable across sectors. Existing coupled analyses (such as by Harrison et al., 2018; or Monier et al., 2018) employ simplified representations of sectoral impacts and do not represent the impacts of extreme events. Thus, while we agree with the reviewer on what would be a desirable future development, we believe that at this point it cannot be achieved by a single study, but will require further efforts by the global change research community. We believe that our present study will be a useful resource for those efforts.

"I am also surprised by the authors' health analysis, the model is extremely simplistic. I am not an expert in that field, but I surprised that there is nothing better than a simple exponential relationship between temperature and mortality?"

The statistical exposure-response functions applied in our study are indeed the state of the art for modelling temperature-mortality relationships. They are based upon established epidemiological methods (Baccini et al., 2008; Gosling et al., 2017; Mitchell et al., 2016). Our aim here is to evaluate the models that are being routinely applied in current climate impact research.

"In summary, there is some useful research here about the comparative capability of modeling to represent the effects of extremes in important physical dimensions. However, as I noted at the outset, I find the paper is substantially over-selling what is being done – the authors are not simulating the "compound" impacts of climate extremes. Rather, the paper is reporting on the capability of a set of biophysical models to predict a set of impact indicators in response to an historical extreme event. Thus, I conclude the paper should not be published. That being said, I think the material in this paper does deserve eventual publication in some form, appropriately presented."

We regret that our usage of the term "compound impacts" is ambiguous and that we did not lay out properly the motivation, aims, and significance of our study. As outlined above, we have extensively revised our manuscript, including removal of the term "compound impacts", and believe that it now becomes clear that our study is not meant to be an integrated assessment, but a novel evaluation of the biophysical models that provide the foundation for integrated assessments (economic or other) of climate change impacts.

For the authors: J. Schewe, September 2018

References

- Arnell, N. W., Lowe, J. A., Brown, S., Gosling, S. N., Gottschalk, P., Hinkel, J., ... Warren, R. F. (2013). A global assessment of the effects of climate policy on the impacts of climate change. *Nature Climate Change*, 3(5), 512–519. <http://doi.org/10.1038/nclimate1793>
- Baccini, M., Biggeri, A., Accetta, G., Kosatsky, T., Katsouyanni, K., Analitis, A., ... Michelozzi, P. (2008). Heat Effects on Mortality in 15 European Cities. *Epidemiology*, 19(5), 711–719. <http://doi.org/10.1097/EDE.0b013e318176bfcd>
- Beck, H. E., van Dijk, A. I. J. M., de Roo, A., Dutra, E., Fink, G., Orth, R., & Schellekens, J. (2016). Global evaluation of runoff from ten state-of-the-art hydrological models. *Hydrology and Earth System Sciences Discussions*, 0, 1–33. <http://doi.org/10.5194/hess-2016-124>
- Borrell, C., Marí-Dell'Olmo, M., Rodríguez-Sanz, M., Garcia-Olalla, P., Caylà, J. A., Benach, J., & Muntaner, C. (2006). Socioeconomic position and excess mortality during the heat wave of 2003 in Barcelona. *European Journal of Epidemiology*, 21(9), 633–640. <http://doi.org/10.1007/s10654-006-9047-4>
- Caminade, C., Kovats, S., Rocklov, J., Tompkins, a. M., Morse, a. P., Colon-Gonzalez, F. J., ... Lloyd, S. J. (2014). Impact of climate change on global malaria distribution. *Proceedings of the National Academy of Sciences*, 1–6. <http://doi.org/10.1073/pnas.1302089111>
- Ciais, P., Reichstein, M., Viovy, N., Granier, A., Ogée, J., Allard, V., ... Valentini, R. (2005). Europe-wide reduction in primary productivity caused by the heat and drought in 2003. *Nature*, 437(7058), 529–33. <http://doi.org/10.1038/nature03972>
- Copa-Cogeca. (2004). *Assessment of the impact of the heat wave and drought of the summer 2003 on agriculture and forestry*. Retrieved from http://docs.gip-ecofor.org/libre/COPA_COGECA_2004.pdf
- Dosio, A., Mentaschi, L., Fischer, E. M., & Wyser, K. (2018). Extreme heat waves under 1.5 °C and 2 °C global warming. *Environmental Research Letters*, 13(5), 054006. <http://doi.org/10.1088/1748-9326/aab827>
- Frank, D., Reichstein, M., Bahn, M., Thonicke, K., Frank, D., Mahecha, M. D., ... Zscheischler, J. (2015). Effects of climate extremes on the terrestrial carbon cycle: concepts, processes and potential future impacts. *Global Change Biology*, 21(8), 2861–2880. <http://doi.org/10.1111/gcb.12916>
- Franzke, C. L. E. (2017). Impacts of a Changing Climate on Economic Damages and Insurance. *Economics of Disasters and Climate Change*, 1(1), 95–110. <http://doi.org/10.1007/s41885-017-0004-3>
- Frieler, K., Levermann, A., Elliott, J., Heinke, J., Arneth, A., Bierkens, M. F. P. F. P., ... Schellnhuber, H. J. J. (2015). A framework for the cross-sectoral integration of multi-model impact projections: land use decisions under climate impacts uncertainties. *Earth System Dynamics*, 6(2), 447–460. <http://doi.org/10.5194/esd-6-447-2015>
- Friend, A. D., Lucht, W., Rademacher, T. T., Keribin, R., Betts, R., Cadule, P., ... Woodward, F. I. (2013). Carbon residence time dominates uncertainty in terrestrial vegetation responses to future climate and atmospheric CO₂. *Proceedings of the National Academy of Sciences of the United States of America*, 2–7. <http://doi.org/10.1073/pnas.1222477110>
- García-Herrera, R., Díaz, J., Trigo, R. M., Luterbacher, J., & Fischer, E. M. (2010). A Review of the European Summer Heat Wave of 2003. *Critical Reviews in Environmental Science and Technology*,

40(4), 267–306. <http://doi.org/10.1080/10643380802238137>

- Gasparri, A., & Armstrong, B. (2011). The Impact of Heat Waves on Mortality. *Epidemiology*, 22(1), 68–73. <http://doi.org/10.1097/EDE.0b013e3181fdcd99>
- Gosling, S. N., Hondula, D. M., Bunker, A., Ibarreta, D., Liu, J., Zhang, X., & Sauerborn, R. (2017). Adaptation to Climate Change: A Comparative Analysis of Modeling Methods for Heat-Related Mortality. *Environmental Health Perspectives*, 125(8), 1–45. <http://doi.org/10.1289/EHP634>
- Gosling, S. N., Lowe, J. A., McGregor, G. R., Pelling, M., & Malamud, B. D. (2009). Associations between elevated atmospheric temperature and human mortality: a critical review of the literature. *Climatic Change*, 92(3–4), 299–341. <http://doi.org/10.1007/s10584-008-9441-x>
- Grize, L., Huss, A., Thommen, O., Schindler, C., & Braun-Fahrlander, C. (2005). Heat wave 2003 and mortality in Switzerland. *Swiss Medical Weekly*, 135(13–14), 200–205.
- Hajat, S., Armstrong, B., Baccini, M., Biggeri, A., Bisanti, L., Russo, A., ... Kosatsky, T. (2006). Impact of High Temperatures on Mortality. *Epidemiology*, 17(6), 632–638. <http://doi.org/10.1097/01.ede.0000239688.70829.63>
- Harrison, P. A., Dunford, R. W., Holman, I. P., Cojocaru, G., Madsen, M. S., Chen, P.-Y., ... Sandars, D. (2018). Differences between low-end and high-end climate change impacts in Europe across multiple sectors. *Regional Environmental Change*. <http://doi.org/10.1007/s10113-018-1352-4>
- Hattermann, F. F., Krysanova, V., Gosling, S. N., Dankers, R., Daggupati, P., Donnelly, C., ... Samaniego, L. (2017). Cross-scale intercomparison of climate change impacts simulated by regional and global hydrological models in eleven large river basins. *Climatic Change*, 141(3), 561–576. <http://doi.org/10.1007/s10584-016-1829-4>
- Hsiang, S., Kopp, R., Jina, A., Rising, J., Delgado, M., Mohan, S., ... Houser, T. (2017). Estimating economic damage from climate change in the United States. *Science*, 356(6345). Retrieved from <http://science.sciencemag.org/content/356/6345/1362/tab-pdf>
- Huang, S., Kumar, R., Flörke, M., Yang, T., Hundecha, Y., Kraft, P., ... Krysanova, V. (2017). Evaluation of an ensemble of regional hydrological models in 12 large-scale river basins worldwide. *Climatic Change*, 141(3), 381–397. <http://doi.org/10.1007/s10584-016-1841-8>
- Hurrell, J. W., Holland, M. M., Gent, P. R., Ghan, S., Kay, J. E., Kushner, P. J., ... Marshall, S. (2013). The Community Earth System Model: A Framework for Collaborative Research. *Bulletin of the American Meteorological Society*, 94(9), 1339–1360. <http://doi.org/10.1175/BAMS-D-12-00121.1>
- Ito, A., Nishina, K., Reyer, C. P. O., François, L., Henrot, A.-J., Munhoven, G., ... Zhao, F. (2017). Photosynthetic productivity and its efficiencies in ISIMIP2a biome models: benchmarking for impact assessment studies. *Environmental Research Letters*, 12(8), 085001. <http://doi.org/10.1088/1748-9326/aa7a19>
- Johnson, H., Kovats, R. S., McGregor, G., Stedman, J., Gibbs, M., Walton, H., ... Black, E. (2005). The impact of the 2003 heat wave on mortality and hospital admissions in England. *Health Statistics Quarterly*, (25), 6–11.
- Le Tertre, A., Lefranc, A., Eilstein, D., Declercq, C., Medina, S., Blanchard, M., ... Ledrans, M. (2006). Impact of the 2003 Heatwave on All-Cause Mortality in 9 French Cities. *Epidemiology*, 17(1), 75–79.

<http://doi.org/10.1097/01.ede.0000187650.36636.1f>

- Leonard, M., Westra, S., Phatak, A., Lambert, M., van den Hurk, B., McInnes, K., ... Stafford-Smith, M. (2014). A compound event framework for understanding extreme impacts. *Wiley Interdisciplinary Reviews: Climate Change*, *5*(1), 113–128. <http://doi.org/10.1002/wcc.252>
- Martínez Navarro, F., Simón-Soria, F., & López-Abente, G. (2004). Valoración del impacto de la ola de calor del verano de 2003 sobre la mortalidad. *Gaceta Sanitaria*, *18*(Supl.1), 250–258. <http://doi.org/10.1157/13062535>
- Michelozzi, P., de Donato, F., Bisanti, L., Russo, A., Cadum, E., DeMaria, M., ... Perucci, C. A. (2005). The impact of the summer 2003 heat waves on mortality in four Italian cities. *Euro Surveillance : Bulletin Europeen Sur Les Maladies Transmissibles = European Communicable Disease Bulletin*, *10*(7), 161–5. Retrieved from <http://www.ncbi.nlm.nih.gov/pubmed/16088045>
- Michelozzi, P., De Sario, M., Accetta, G., De'Donato, F., Kirchmayer, U., D'Ovidio, M., & Perucci, C. A. (2006). Temperature and summer mortality: geographical and temporal variations in four Italian cities. *Journal of Epidemiology and Community Health*, *60*, 417–423. <http://doi.org/10.1136/jech.2005.040857>
- Milly, P. C. D., & Dunne, K. A. (2017). A Hydrologic Drying Bias in Water-Resource Impact Analyses of Anthropogenic Climate Change. *JAWRA Journal of the American Water Resources Association*. <http://doi.org/10.1111/1752-1688.12538>
- Mitchell, D., Heaviside, C., Vardoulakis, S., Huntingford, C., Masato, G., P Guillod, B., ... Allen, M. (2016). Attributing human mortality during extreme heat waves to anthropogenic climate change. *Environmental Research Letters*, *11*(7), 074006. <http://doi.org/10.1088/1748-9326/11/7/074006>
- Monier, E., Paltsev, S., Sokolov, A., Chen, Y.-H. H., Gao, X., Ejaz, Q., ... Haigh, M. (2018). Toward a consistent modeling framework to assess multi-sectoral climate impacts. *Nature Communications*, *9*(1), 660. <http://doi.org/10.1038/s41467-018-02984-9>
- Müller, C., Elliott, J., Chryssanthacopoulos, J., Arneth, A., Balkovic, J., Ciais, P., ... Yang, H. (2017). Global gridded crop model evaluation: benchmarking, skills, deficiencies and implications. *Geoscientific Model Development*, *10*(4), 1403–1422. <http://doi.org/10.5194/gmd-10-1403-2017>
- Orth, R., Zscheischler, J., Seneviratne, S. I., Schär, C., Barriopedro, D., Fischer, E. M., ... Jones, P. D. (2016). Record dry summer in 2015 challenges precipitation projections in Central Europe. *Scientific Reports*, *6*, 28334. <http://doi.org/10.1038/srep28334>
- Pecl, G. T., Araújo, M. B., Bell, J. D., Blanchard, J., Bonebrake, T. C., Chen, I.-C., ... Williams, S. E. (2017). Biodiversity redistribution under climate change: Impacts on ecosystems and human well-being. *Science*, *355*(6332), eaai9214. <http://doi.org/10.1126/science.aai9214>
- Pindyck, R. S. (2013). Climate Change Policy: What Do the Models Tell Us? *Journal of Economic Literature*, *51*(3), 860–872. <http://doi.org/10.1257/jel.51.3.860>
- Piontek, F., Müller, C., Pugh, T. A. M., Clark, D. B., Deryng, D., Elliott, J., ... Schellnhuber, H. J. (2014). Multisectoral climate impact hotspots in a warming world. *Proceedings of the National Academy of Sciences of the United States of America*, *111*(9). <http://doi.org/10.1073/pnas.1222471110>
- Reichstein, M., Ciais, P., Papale, D., Valentini, R., Running, S., Viovy, N., ... Zhao, M. (2007). Reduction of

- ecosystem productivity and respiration during the European summer 2003 climate anomaly: a joint flux tower, remote sensing and modelling analysis. *Global Change Biology*, 13(3), 634–651.
<http://doi.org/10.1111/j.1365-2486.2006.01224.x>
- Reyer, C. P. O., Bathgate, S., Blennow, K., Borges, J. G., Bugmann, H., Delzon, S., ... Hanewinkel, M. (2017). Are forest disturbances amplifying or canceling out climate change-induced productivity changes in European forests? *Environmental Research Letters*, 12(3), 034027.
<http://doi.org/10.1088/1748-9326/aa5ef1>
- Rosenzweig, C., Elliott, J., Deryng, D., Ruane, A. C., Müller, C., Arneth, A., ... Jones, J. W. (2014). Assessing agricultural risks of climate change in the 21st century in a global gridded crop model intercomparison. *Proceedings of the National Academy of Science*, (14), 1–6.
<http://doi.org/10.1073/pnas.1222463110>
- Running, S. W., & Zhao, M. (2015). Daily GPP and annual NPP (MOD17A2/A3) products NASA Earth Observing System MODIS land algorithm. *MOD17 User's Guide*. Retrieved from https://www.nts.gov/files/modis/MOD17UsersGuide2015_v3.pdf
- Schär, C., Vidale, P. L., Lüthi, D., Frei, C., Häberli, C., Liniger, M. A., & Appenzeller, C. (2004). The role of increasing temperature variability in European summer heatwaves. *Nature*, 427(6972), 332–6.
<http://doi.org/10.1038/nature02300>
- Schauberger, B., Archontoulis, S., Arneth, A., Balkovic, J., Ciais, P., Deryng, D., ... Frieler, K. (2017). Consistent negative response of US crops to high temperatures in observations and crop models. *Nature Communications*, 8, 13931. <http://doi.org/10.1038/ncomms13931>
- Schewe, J., Heinke, J., Gerten, D., Haddeland, I., Arnell, N. W., Clark, D. B., ... Kabat, P. (2014). Multimodel assessment of water scarcity under climate change. *Proceedings of the National Academy of Sciences of the United States of America*, 111(9). <http://doi.org/10.1073/pnas.1222460110>
- Tobías, A., de Olalla, P. G., Linares, C., Bleda, M. J., Caylà, J. A., & Díaz, J. (2010). Short-term effects of extreme hot summer temperatures on total daily mortality in Barcelona, Spain. *International Journal of Biometeorology*, 54(2), 115–117. <http://doi.org/10.1007/s00484-009-0266-8>
- Traore, A. K., Ciais, P., Vuichard, N., Poulter, B., Viovy, N., Guimberteau, M., ... Fisher, J. B. (2014). Evaluation of the ORCHIDEE ecosystem model over Africa against 25 years of satellite-based water and carbon measurements. *Journal of Geophysical Research: Biogeosciences*, 119(8), 1554–1575.
<http://doi.org/10.1002/2014JG002638>
- van der Velde, M., Wriedt, G., & Bouraoui, F. (2010). Estimating irrigation use and effects on maize yield during the 2003 heatwave in France. *Agriculture, Ecosystems & Environment*, 135(1–2), 90–97.
<http://doi.org/10.1016/j.agee.2009.08.017>
- van Vliet, M. T. H., van Beek, L. P. H., Eisner, S., Flörke, M., Wada, Y., & Bierkens, M. F. P. (2016). Multi-model assessment of global hydropower and cooling water discharge potential under climate change. *Global Environmental Change*, 40, 156–170.
<http://doi.org/10.1016/j.gloenvcha.2016.07.007>
- van Vliet, M. T. H., Wiberg, D., Leduc, S., & Riahi, K. (2016). Power-generation system vulnerability and adaptation to changes in climate and water resources. *Nature Climate Change*, 6(4), 375–380.
<http://doi.org/10.1038/nclimate2903>

Whitmee, S., Haines, A., Beyrer, C., Boltz, F., Capon, A. G., de Souza Dias, B. F., ... Yach, D. (2015). Safeguarding human health in the Anthropocene epoch: report of The Rockefeller Foundation–Lancet Commission on planetary health. *The Lancet*, 386(10007), 1973–2028.
[http://doi.org/10.1016/S0140-6736\(15\)60901-1](http://doi.org/10.1016/S0140-6736(15)60901-1)

Zaherpour, J., Gosling, S. N., Mount, N., Müller Schmied, H., Veldkamp, T. I. E., Dankers, R., ... Wada, Y. (2018). Worldwide evaluation of mean and extreme runoff from six global-scale hydrological models that account for human impacts. *Environmental Research Letters*.
<http://doi.org/10.1088/1748-9326/aac547>

Reviewer #1 (Remarks to the Author):

The authors have addressed all my prior comments. While we can continue iterating and fine tuning, I believe the point of diminishing marginal returns has been reached. This article is exciting and deserves to be published in a timely manner. I would recommend publication of this revised article as is.

Auroop R. Ganguly